

# Inner products in integrable Richardson-Gaudin models

**Pieter W. Claeys[1,2,3⋆], Dimitri van Neck[2,3] and Stijn de Baerdemacker[2,3,4]**

**1** Institute for Theoretical Physics, University of Amsterdam, Science Park 904,
1098 XH Amsterdam, The Netherlands
**2** Department of Physics and Astronomy, Ghent University, Proeftuinstraat 86,
9000 Ghent, Belgium
**3** Center for Molecular Modeling, Ghent University, Technologiepark 903,
9052 Ghent, Belgium
**4** Department of Inorganic and Physical Chemistry, Ghent University,
Krijgslaan 281 (S3), 9000 Ghent, Belgium

⋆ PieterW.Claeys@UGent.be

## Abstract

We present the inner products of eigenstates in integrable Richardson-Gaudin models from two different perspectives and derive two classes of Gaudin-like determinant expressions for such inner products. The requirement that one of the states is on-shell arises naturally by demanding that a state has a dual representation. By implicitly combining these different representations, inner products can be recast as domain wall boundary partition functions. The structure of all involved matrices in terms of Cauchy matrices is made explicit and used to show how one of the classes returns the Slavnov determinant formula.

Furthermore, this framework provides a further connection between two different approaches for integrable models, one in which everything is expressed in terms of rapidities satisfying Bethe equations, and one in which everything is expressed in terms of the eigenvalues of conserved charges, satisfying quadratic equations.


# 1 Introduction

Integrable models have proven to be a powerful tool in the study of many-body physics. These models are often characterized by two key properties - they support a large amount of conserved quantities, and they can be solved exactly using Bethe ansatz methods [1, 2]. The eigenstates of integrable models are given by Bethe states, which are characterized by a set of variables, so-called rapidities or Bethe roots, which have to obey a set of coupled non-linear equations, the Bethe equations. The underlying framework of integrability then often allows for the exact calculation of physical observables from these eigenstates. For example correlation coefficients, one of the main means of extracting the underlying physics from a given wave function, can be efficiently calculated in these models [2]. More specifically, these correlation coefficients can often be reduced to (polynomial sums of) determinants. Since determinants can be efficiently evaluated numerically, such expressions have subsequently allowed for the investigation of various large-scale integrable systems in many different contexts, where e.g. their appearance in quantum quenches has attracted a lot of attention [3, 4]. Here, one of the key expressions is the well-known Slavnov formula for the inner product between two Bethe states, one with rapidities satisfying the Bethe equations (an on-shell state), and one with arbitrary rapidities (an off-shell state) [5]. Such determinant expressions provide a basic building block for the calculation of correlation coefficients from the Bethe states, which has allowed for massive simplifications in the calculations of correlation coefficients in these models [6–10]. Such inner products can also be interpreted as domain wall boundary partition functions (DWPF), which can similarly be expressed as determinants [11–14].

In the following, we investigate the structure of these inner products in Richardson-Gaudin

models [5,7,15–18]. The existence of two distinct representations for each on-shell Bethe state allows such inner products to be recast as DWPFs, and we show here how the Slavnov determinant follows as a corollary. The demand that one of the two states is on-shell, a prerequisite for the existence of such determinant expressions, then follows naturally by requiring that a so-called dual representation of one of the two states exist, which is a known characteristic of on-shell states. In the (numerical) treatment of Richardson-Gaudin models, two separate approaches exist. In the first, on-shell Bethe states are obtained by solving the nonlinear coupled Richardson-Gaudin (or Bethe) equations for the *rapidities*, after which correlation coefficients are calculated from these rapidities using the Slavnov determinant. In the second approach, the so-called *eigenvalue-based* approach, eigenstates are characterized by solving for a set of variables determining the eigenvalues of the conserved charges for each eigenstate. The correlation coefficients can then also be calculated as DWPFs depending only on these new variables, keeping the rapidities implicit [19–22]. In this work, the connection between these approaches is made explicit, highlighting the Cauchy structure of all involved matrices, which follows from the rational functions defining these models.

Recently, Richardson-Gaudin integrable models where no clear reference state exists have been under much attention [23–26]. While complicating the structure of the Bethe equations, this only has minor influence on the eigenvalue-based equations. This also allows for a straightforward derivation of eigenvalue-based determinant expressions for inner products in these models. While it would be worthwhile to obtain similar connections as presented in this work for these models, we restrict ourselves to the better-known class of models containing a clear reference state.

The paper is organized as follows. In section 2, we give an overview of the theory of Richardson-Gaudin models, after which we give an overview of the main results of this work in section 3. The connection with previously known expressions is then made explicit in sections 4 and 5, where we first derive several crucial properties of Cauchy matrices, which allow for the use of dual representations and DWPFs. For the clarity of presentation, we restrict ourselves to the so-called rational model throughout the paper, and show in section 6 how this can be straightforwardly extended to hyperbolic models. Section 7 is then reserved for conclusions.

# 2 Richardson-Gaudin models

## 2.1 Generalized Gaudin algebra

In this section, we will give a brief overview of the theory of Richardson-Gaudin systems from the viewpoint of generalized Gaudin algebras, following the presentation from [27]. It is worth mentioning that the integrability of these models can also be derived within the framework of the Algebraic Bethe Ansatz through taking the so-called quasi-classical limit (see [8, 18, 28–32]).

As mentioned in the introduction, integrable models are characterized by two features going hand in hand - the existence of a large set of conserved quantities, and their exact solvability by Bethe ansatz. These both follow naturally from the concept of a Generalized Gaudin algebra (GGA). Such an algebra is defined by operators $S^x(u), S^y(u), S^z(u)$ satisfying

the commutation relations

$$[S^x(u), S^y(v)] = \frac{1}{u-v}(S^z(u) - S^z(v)), \tag{1}$$

$$[S^y(u), S^z(v)] = \frac{1}{u-v}(S^x(u) - S^x(v)), \tag{2}$$

$$[S^z(u), S^x(v)] = \frac{1}{u-v}(S^y(u) - S^y(v)), \tag{3}$$

$$[S^\kappa(u), S^\kappa(v)] = 0, \qquad \kappa = x, y, z, \tag{4}$$

with $u, v \in \mathbb{C}$. This is an infinite dimensional Lie algebra, highly reminiscent of the $su(2)$ algebra. Extending the similarity to $su(2)$, raising and lowering operators can be defined as

$$S^+(u) = S^x(u) + iS^y(u), \qquad S^-(u) = S^x(u) - iS^y(u), \tag{5}$$

which satisfy the commmutation relations

$$\left[S^+(u), S^-(v)\right] = 2\frac{S^z(u) - S^z(v)}{u-v}, \tag{6}$$

$$\left[S^z(u), S^\pm(v)\right] = \pm\frac{S^\pm(u) - S^\pm(v)}{u-v}. \tag{7}$$

Given such an algebra, a continuous family of mutually commuting operators[1] can be defined as

$$\mathbb{S}^2(u) = S^x(u)^2 + S^y(u)^2 + S^z(u)^2$$
$$= \frac{1}{2}\left(S^+(u)S^-(u) + S^-(u)S^+(u)\right) + S^z(u)^2. \tag{8}$$

Note that although these resemble the Casimir operator of $su(2)$, they do not act as Casimir operators for the GGA since they do not commute with its generators. It follows from the given commutation relations of the GGA that

$$[\mathbb{S}^2(u), \mathbb{S}^2(v)] = 0, \quad \forall u, v \in \mathbb{C}, \tag{9}$$

so these operators generate a continuous set of commuting operators, leading to a continuous set of conserved charges.

Their exact solvability also follows from the definition of the GGA. Bethe ansatz states can be constructed as

$$|v_1 \ldots v_N\rangle = S^+(v_1) \ldots S^+(v_N)|0\rangle = \prod_{a=1}^{N} S^+(v_a)|0\rangle, \tag{10}$$

containing a product of raising operators acting on a vacuum reference state $|0\rangle$, which is defined as being both annihilated by the lowering operator $S^-(u)|0\rangle = 0, \forall\, u \in \mathbb{C}$, and being an eigenstate of $S^z(u), \forall\, u \in \mathbb{C}$. This wave function then depends on a set of free parameters $\{v_a\} = \{v_1 \ldots v_N\}$, so-called rapidities or Bethe roots.

Using the commutation relations, the action of $\mathbb{S}^2(u)$ on such a Bethe state can be shown to be

$$\mathbb{S}^2(u)|v_1 \ldots v_N\rangle = \left(F(u) + \sum_{a=1}^{N}\left[-2\frac{F(v_a)}{u-v_a} + \sum_{b\neq a}^{N}\frac{1}{u-v_a}\frac{1}{u-v_b}\right]\right)|v_1 \ldots v_N\rangle$$
$$+ \sum_{a=1}^{N}\frac{2}{u-v_a}\left[F(v_a) - \sum_{b\neq a}^{N}\frac{1}{v_a-v_b}\right]|v_1 \ldots v_a \to u \ldots v_N\rangle, \tag{11}$$

---

[1]These provide the GGA equivalent of the transfer matrix in the Algebraic Bethe Ansatz [2, 7, 33].

where the state $|v_1 \ldots v_a \to u \ldots v_N\rangle$ is built as in Eq. (10), but with a single rapidity replaced by $u$, and $F(u)$ is defined as $S^z(u)|0\rangle = F(u)|0\rangle$. The action of $\mathbb{S}^2(u)$ gives rise to two contributions: one diagonal part and a set of $N$ off-diagonal (unwanted) states. So far, the Bethe state has been defined in terms of $N$ free rapidities. If these are chosen in such a way that the unwanted parts vanish, the resulting Bethe state is an eigenstate by construction. Indeed, if the rapidities satisfy the Richardson-Gaudin (or Bethe) equations

$$F(v_a) - \sum_{b \neq a}^{N} \frac{1}{v_a - v_b} = 0, \qquad \text{with} \qquad S^z(v_a)|0\rangle = F(v_a)|0\rangle, \qquad a = 1 \ldots N, \tag{12}$$

the resulting Bethe state is an eigenstate.

A standard realization of this algebra exists within the context of spin-1/2 models, leading to both the Richardson [15, 16] and the Gaudin models [17]. For a set of $L$ $su(2)$-algebras $\{S_i^z, S_i^+, S_i^-\}$, labelled $i = 1 \ldots L$ and satisfying

$$[S_i^+, S_j^-] = 2\delta_{ij}S_i^z, \qquad [S_i^z, S_j^\pm] = \pm\delta_{ij}S_i^\pm, \tag{13}$$

a GGA can be realized by the following operators

$$S^z(u) = -\frac{1}{g} - \sum_{i=1}^{L} \frac{S_i^z}{u - \epsilon_i}, \qquad S^+(u) = \sum_{i=1}^{L} \frac{S_i^+}{u - \epsilon_i}, \qquad S^-(u) = \sum_{i=1}^{L} \frac{S_i^-}{u - \epsilon_i}, \tag{14}$$

and $|0\rangle = |\downarrow \ldots \downarrow\rangle$. Both $g$ and $\{\epsilon_1 \ldots \epsilon_L\}$ are free variables, with the latter playing the role of the inhomogeneities in the usual transfer matrix construction.

The resulting $\mathbb{S}^2(u)$ has single poles in $u = \epsilon_i$, $i = 1 \ldots L$ and can be expanded as

$$\mathbb{S}^2(u) = \frac{2}{g} \sum_{i=1}^{L} \frac{R_i}{u - \epsilon_i} + \langle 0|\mathbb{S}^2(u)|0\rangle, \tag{15}$$

where the residues lead to a set of commuting operators

$$R_i = \left(S_i^z + \frac{1}{2}\right) + g \sum_{j \neq i}^{L} \frac{1}{\epsilon_i - \epsilon_j}\left[\frac{1}{2}\left(S_i^+ S_j^- + S_i^- S_j^+\right) + \left(S_i^z S_j^z - \frac{1}{4}\right)\right], \qquad i = 1 \ldots L. \tag{16}$$

The common eigenstates of these operators now follow from the GGA as

$$|v_1 \ldots v_N\rangle = \prod_{a=1}^{N}\left(\sum_{i=1}^{L} \frac{S_i^+}{\epsilon_i - v_a}\right)|\downarrow \ldots \downarrow\rangle, \tag{17}$$

described by rapidities $\{v_1 \ldots v_N\}$ satisfying the equations

$$\frac{1}{g} + \frac{1}{2}\sum_{i=1}^{L} \frac{1}{\epsilon_i - v_a} - \sum_{b \neq a}^{N} \frac{1}{v_b - v_a} = 0, \qquad a = 1 \ldots N. \tag{18}$$

The action of the conserved charges on these states can then be obtained from the eigenvalues of $\mathbb{S}^2(u)$ as

$$R_i|v_1 \ldots v_N\rangle = -\frac{g}{2}\left[\sum_{a=1}^{N} \frac{1}{\epsilon_i - v_a}\right]|v_1 \ldots v_N\rangle$$

$$= -\frac{g}{2}\Lambda_i(\{v_a\})|v_1 \ldots v_N\rangle. \tag{19}$$

Here, $\Lambda_i(\{v_a\}) = \sum_{a=1}^{N} \frac{1}{\epsilon_i - v_a}$ encodes the dependence of the eigenvalues of the conserved charges on the rapidities, and is therefore also known as an eigenvalue-based variable [34].

## 2.2 Eigenvalue-based framework

Conventionally, solving integrable systems consists of first solving the Bethe equations (18) for the rapidities, which parametrize the eigenstates, and then extracting correlation coefficients from these rapidities using determinant expressions following from Slavnov's determinant. An alternative approach avoids the explicit use of rapidities. Instead, it can be shown that the conserved charges (16) satisfy a set of quadratic equations [24, 35]

$$R_i^2 = R_i - \frac{g}{2} \sum_{j \neq i}^{L} \frac{R_i - R_j}{\epsilon_i - \epsilon_j}, \qquad i = 1 \ldots L. \tag{20}$$

Because these operators commute by construction, they can be diagonalized simultaneously, and this equation can be rewritten at the level of the eigenvalues. This leads to a set of quadratic equations for the eigenvalue-based variables

$$\Lambda_i^2 + \frac{2}{g} \Lambda_i - \sum_{j \neq i}^{L} \frac{\Lambda_i - \Lambda_j}{\epsilon_i - \epsilon_j} = 0, \qquad i = 1 \ldots L, \tag{21}$$

with

$$\Lambda_i \equiv \Lambda_i(\{\nu_a\}) = \sum_{a=1}^{N} \frac{1}{\epsilon_i - \nu_a}, \qquad i = 1 \ldots L. \tag{22}$$

These equations are also known as the substituted or quadratic Bethe equations. Originally obtained by Babelon and Talalaev [19], these were later extended in a series of articles towards all known Richardson-Gaudin models [21,22,26,34,36]. They arose as a way of circumventing the singular behaviour of the regular Richardson-Gaudin equations, since they do not exhibit the singularities associated with the poles in the regular Bethe equations [37–39]. In many ways, this framework is similar to the use of Baxter's $T - Q$-relation in the Algebraic Bethe Ansatz in immediately solving for the eigenvalues of the conserved charges [40].

Remarkably, it is possible to obtain expressions for inner products between two Bethe states which only depend explicitly on these new variables, further cementing their usefulness, so it is not always necessary to extract the rapidities from these variables [12–14, 21, 22].

## 2.3 Moving between both frameworks

Nevertheless, not all possible correlation coefficients or matrix elements can be expressed in terms of the eigenvalue-based variables, so it is necessary to obtain an efficient way of moving between representations (rapidity-based and eigenvalue-based). From a numerical point of view these different representations presents a trade-off. On the one hand, compared to the Richardson-Gaudin equations the eigenvalue-based equations can be solved in a much more straightforward way, but it is then not trivial to obtain the rapidities from the eigenvalue-based variables. On the other hand, while directly solving the Bethe equations (18) for the rapidities is a more involved task, the eigenvalue-based variables can afterwards be immediately obtained by simply plugging the rapidities into Eq. (22).

One way of (numerically) extracting the rapidities from the eigenvalue-based variables and inverting Eq. (22) is by defining a polynomial with the rapidities as roots

$$P(z) = \prod_{a=1}^{N} (z - \nu_a), \tag{23}$$

which satisfies the ordinary differential equation [35, 36, 41]

$$P''(z) + \left[ \frac{2}{g} + \sum_{i=1}^{L} \frac{1}{\epsilon_i - z} \right] P'(z) - \left[ \sum_{i=1}^{L} \frac{\Lambda_i}{\epsilon_i - z} \right] P(z) = 0. \tag{24}$$

Once the variables $\Lambda_i$ are known, this differential equation is well-defined, and efficient algorithms exist for finding the roots of a polynomial satisfying such a differential equation [36]. Remarkably, the differential equation (24) can be derived starting from either the Bethe equations (18) or the eigenvalue-based equations (21), and can be considered the connection between both frameworks [35].

This has profound consequences, as recently pointed out in [35]. Since the eigenvalue-based equations follow from operator identities, they are necessarily complete, and the equivalence between the Bethe equations and the eigenvalue-based equations then implies the completeness of the Bethe equations and the associated Bethe ansatz states.

Given two different ways of solving the Bethe equations, the differential equation (24) then poses another way of solving for the eigenstates, combining both approaches. Instead of first solving for the eigenvalue-based variables and afterwards extracting the rapidities, the expansion of the eigenvalue-based variables in terms of the rapidities can be introduced, and the full equation can be solved for the rapidities in a self-consistent way, as done in [42]. The correspondence between Bethe equations and differential equations has similarly led to a plethora of techniques for solving the Bethe equations [42–49].

# 3 Overview of the main results

## 3.1 Inner products

The main object of interest in this work is the inner product of Bethe states. In this section, a series of determinant expressions for such inner products will be presented without proof, after which the mathematical identities and reasoning underlying these results will be presented in later sections.

Suppose we have two sets of rapidities, one ($\{v_a\} = \{v_1 \dots v_N\}$) obtained by solving the Richardson-Gaudin equations, and another ($\{w_b\} = \{w_1 \dots w_N\}$) given by arbitrary complex variables. If the rapidities satisfy the Richardson-Gaudin equations, the resulting Bethe state is also known as an *on-shell* state, whereas a Bethe state defined by arbitrary variables is known as an *off-shell* state. The inner product between an on-shell and an off-shell state is then famously given by Slavnov's determinant formula [2, 5, 7, 50]

$$\langle v_1 \dots v_N | w_1 \dots w_N \rangle = \frac{\prod_b \prod_{a \neq b}(v_a - w_b)}{\prod_{a<b}(v_b - v_a)\prod_{b<a}(w_b - w_a)} \det S_N(\{v_a\}, \{w_b\}), \qquad (25)$$

with $S_N(\{v_a\}, \{w_b\})$ an $N \times N$ matrix defined as

$$S_N(\{v_a\}, \{w_b\})_{ab} = \frac{v_b - w_b}{v_a - w_b}\left(\sum_{i=1}^{L}\frac{1}{(v_a - \epsilon_i)(w_b - \epsilon_i)} - 2\sum_{c \neq a}^{N}\frac{1}{(v_a - v_c)(w_b - v_c)}\right). \qquad (26)$$

The first main results of this paper are the alternative expressions

$$\langle v_1 \dots v_N | w_1 \dots w_N \rangle = (-1)^N \left(\frac{g}{2}\right)^{L-2N} \det J_L(\{v_a\}, \{w_b\}), \qquad (27)$$

with $J_L(\{v_a\}, \{w_b\})$ an $L \times L$ matrix defined as

$$J_L(\{v_a\}, \{w_b\})_{ij} = \begin{cases} \frac{2}{g} + \Lambda_i(\{v_a\}) + \Lambda_i(\{w_b\}) - \sum_{k \neq i}^{L}\frac{1}{\epsilon_i - \epsilon_k} & \text{if } i = j \\ -\frac{1}{\epsilon_i - \epsilon_j} & \text{if } i \neq j \end{cases}, \qquad (28)$$

where the diagonal elements depend on the eigenvalue-based variables as defined in Eq. (22), and the dependence on the rapidities can be made explicit as

$$J_L(\{v_a\},\{w_b\})_{ij} = \begin{cases} \frac{2}{g} + \sum_{a=1}^{N} \frac{1}{\epsilon_i - v_a} + \sum_{b=1}^{N} \frac{1}{\epsilon_i - w_b} - \sum_{k \neq i}^{L} \frac{1}{\epsilon_i - \epsilon_k} & \text{if } i = j \\ -\frac{1}{\epsilon_i - \epsilon_j} & \text{if } i \neq j \end{cases}, \quad (29)$$

and the related and equivalent determinant expression

$$\langle v_1 \dots v_N | w_1 \dots w_N \rangle = (-1)^N \det K_{2N}(\{v_a\},\{w_b\}), \quad (30)$$

with $K_{2N}(\{v_a\},\{w_b\})$ an $2N \times 2N$ matrix defined in terms of

$$\{x_1 \dots x_{2N}\} = \{v_1 \dots v_N\} \cup \{w_1 \dots w_N\} \quad (31)$$

as

$$K_{2N}(\{v_a\},\{w_b\})_{ab} = \begin{cases} \frac{2}{g} - \sum_{i=1}^{L} \frac{1}{x_a - \epsilon_i} + \sum_{c \neq a}^{2N} \frac{1}{x_a - x_c} & \text{if } a = b \\ -\frac{1}{x_a - x_b} & \text{if } a \neq b \end{cases}. \quad (32)$$

These two expressions are clearly related by exchanging the role of the rapidities and the inhomogeneities, a feature which will be further detailed in following sections. Furthermore, the matrix (28) is well-suited for the eigenvalue-based framework, since it only depends on the rapidities through the eigenvalue-based variables in the diagonal elements. Because of this specific structure, it is not necessary to explicitly know the rapidities in order to evaluate the matrix elements. Note that the diagonal elements are reminiscent of the eigenvalue-based equations (21), a feature which will be further commented on in later sections. Similarly, the matrix (30) only depends on the inhomogeneities through the diagonal elements, in a form similar to the Bethe equations (18). The orthogonality of two different eigenstates can then easily be shown by exploiting the similarity of the diagonal elements to the eigenvalue-based/Bethe equations, as shown in Appendix A.

These matrices also exhibit the same structure as the Gaudin matrix for the normalization of an on-shell state [17], hence the denomination of Gaudin-like matrices. For this kind of matrices, the off-diagonal elements only depend on the difference of two rapidities (inhomogeneities). The diagonal elements then contain two summations, one over all but one rapidities (inhomogeneities), and one over all inhomogeneities (rapidities). Remarkably, these Gaudin-like structures are not limited to Richardson-Gaudin models and have been observed in general integrable models such as the XXZ spin chain [51–54] and the Lieb-Liniger model [55, 56].

The second main result of this paper is the identification of the Cauchy matrix as the fundamental building block underneath all matrix expressions, being directly responsible for the Gaudin-like structure. In the following sections, it will also be shown that the Slavnov determinant (25) can be derived starting from Eq. (30), similarly exposing its structure in terms of Cauchy matrices. We conclude this section with two well-known specific cases of inner products which have special use in calculations, leading to the Gaudin determinant and the Izergin-Borchardt determinant, and show how they fit within this scheme.

## 3.2 Gaudin determinant

If the two sets of rapidities coincide, Slavnov's determinant expression (25) reduces to the Gaudin determinant for the normalization of Bethe states [17], given by

$$\langle v_1 \dots v_N | v_1 \dots v_N \rangle = \det G_N(v_1 \dots v_N), \quad (33)$$

with $G_N(v_1 \dots v_N)$ an $N \times N$ matrix defined as

$$G_N(v_1 \dots v_N)_{ab} = \begin{cases} \sum_{i=1}^{L} \frac{1}{(\epsilon_i - v_a)^2} - 2\sum_{c \neq a}^{N} \frac{1}{(v_c - v_a)^2} & \text{if } a = b \\ \frac{2}{(v_a - v_b)^2} & \text{if } a \neq b \end{cases}. \quad (34)$$

The Gaudin matrix is also obtained by taking the limit of coinciding rapidities for Eq. (30). Using elementary row and column operations, the determinant of the $2N \times 2N$ matrix from Eq. (30) can in this limit be reduced to that of the $N \times N$ Gaudin matrix.

Interestingly, the eigenvalue-based expression for the inner product (29) leads to an alternative expression for the normalization as

$$\langle v_1 \ldots v_N | v_1 \ldots v_N \rangle = (-1)^N \left(\frac{g}{2}\right)^{L-2N} \det J_L(\{v_a\}), \tag{35}$$

with $J_L(\{v_a\})$ an $L \times L$ matrix defined as

$$J_L(\{v_a\})_{ij} = \begin{cases} \frac{2}{g} + 2\Lambda_i(\{v_a\}) - \sum_{k \neq i}^{L} \frac{1}{\epsilon_i - \epsilon_k} & \text{if } i = j \\ -\frac{1}{\epsilon_i - \epsilon_j} & \text{if } i \neq j \end{cases}, \tag{36}$$

$$= \begin{cases} \frac{2}{g} + 2\sum_{a=1}^{N} \frac{1}{\epsilon_i - v_a} - \sum_{k \neq i}^{L} \frac{1}{\epsilon_i - \epsilon_k} & \text{if } i = j \\ -\frac{1}{\epsilon_i - \epsilon_j} & \text{if } i \neq j \end{cases}. \tag{37}$$

It has already been mentioned how the structure of this matrix resembles that of the presented determinant expressions (29) and (30). This similarity and the related use of both determinant expressions can be even further established, since the Gaudin matrix is identical to the Jacobian of the Bethe equations (18), which seems to be a common property of integrable systems [57–62], while the alternative matrix (36) is exactly the Jacobian of the eigenvalue-based equations (21).

## 3.3 Izergin-Borchardt determinant

Another frequently encountered case is where the off-shell state simplifies to a simple product state $|i_1 \ldots i_N\rangle = \prod_{b=1}^{N} S_{i_b}^{\dagger} |\downarrow \ldots \downarrow\rangle$, where Slavnov's determinant expression results in an Izergin-Borchardt determinant [63–65]

$$\langle i_1 \ldots i_N | v_1 \ldots v_N \rangle = \frac{\prod_{a,b}(\epsilon_{i_b} - v_a)}{\prod_{c>b}(\epsilon_{i_b} - \epsilon_{i_c}) \prod_{c<a}(v_a - v_c)} \det\left[\frac{1}{(\epsilon_{i_b} - v_a)^2}\right], \tag{38}$$

which contains the determinant of an $N \times N$ matrix with matrix elements $\frac{1}{(\epsilon_{i_b} - v_a)^2}$. The alternative matrix following from Eq. (29) corresponding to the Izergin-Borchardt determinant then reads

$$\langle \{i_b\} | \{v_a\} \rangle = \det J_N(\{v_a\}, \{i_b\}), \tag{39}$$

with $J_N(\{v_a\}, \{i_b\})$ an $N \times N$ matrix defined as

$$J_N(\{v_a\}, \{i_b\})_{ab} = \begin{cases} \sum_{c=1}^{N} \frac{1}{\epsilon_{i_a} - v_c} - \sum_{c \neq a}^{N} \frac{1}{\epsilon_{i_a} - \epsilon_{i_c}} & \text{if } a = b \\ -\frac{1}{\epsilon_{i_a} - \epsilon_{i_b}} & \text{if } a \neq b \end{cases}, \tag{40}$$

while Eq. (30) gives rise to

$$\langle \{i_b\} | \{v_a\} \rangle = \det K_N(\{v_a\}, \{i_b\}), \tag{41}$$

with $K_N(\{v_a\}, \{i_b\})$ an $N \times N$ matrix defined as

$$K_N(\{v_a\}, \{i_b\})_{ab} = \begin{cases} -\sum_{c=1}^{N} \frac{1}{v_a - \epsilon_{i_c}} + \sum_{c \neq a}^{N} \frac{1}{v_a - v_c} & \text{if } a = b \\ -\frac{1}{v_a - v_b} & \text{if } a \neq b \end{cases}. \tag{42}$$

In fact, these results were originally obtained in [12], and served as the building blocks for this work. We should stress that the matrices in Eqs. (29) and (36) also appeared in [12] where

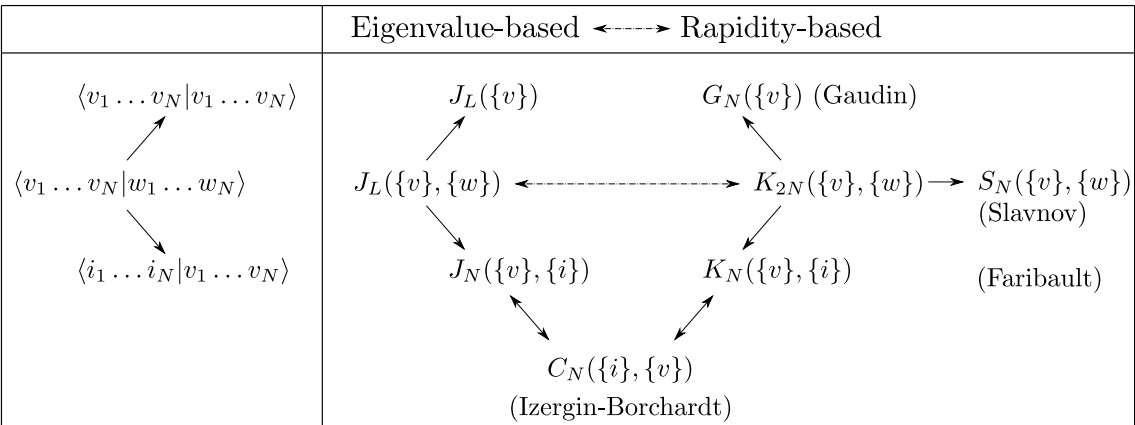

Figure 1: Overview of all relevant matrices and their interconnections for inner products in Richadson-Gaudin models.

it was shown that inner products and normalizations are *proportional* to the determinants of these respective matrices. We show here that the proportionality factor can be immediately evaluated as a single constant, independent of both the rapidities and the inhomogeneities of the model. This will be proven in section 5.

An overview of all relevant matrices and their counterparts in both frameworks is given in Figure 1.

## 4 Properties of Cauchy matrices

In order to derive the previously-presented determinant expressions, the Cauchy structure of all matrices needs to be made clear. Several properties of Cauchy matrices will then be used in order to navigate between different determinant expressions, which will be presented in this section. Necessary proofs have been moved to Appendix B for clarity of presentation.

To set the stage, assume we have two sets of variables $\{\epsilon_1 \dots \epsilon_N\} = \{\epsilon_i\}$ and $\{x_1 \dots x_N\} = \{x_\alpha\}$. Given two such sets, an $N \times N$ Cauchy matrix $C$ is defined by the matrix elements

$$C_{i\alpha} = \frac{1}{\epsilon_i - x_\alpha}. \tag{43}$$

The inverse of this matrix is related to the transpose of this matrix through two diagonal matrices [66], which are defined in terms of two polynomials $p(x) = \prod_i (x - \epsilon_i)$ and $q(x) = \prod_\alpha (x - x_\alpha)$ as

$$(D_\epsilon)_{ii} = \frac{q(\epsilon_i)}{p'(\epsilon_i)}, \qquad (D_x)_{\alpha\alpha} = \frac{p(x_\alpha)}{q'(x_\alpha)}, \tag{44}$$

such that

$$C^{-1} = -D_x C^T D_\epsilon. \tag{45}$$

Permanents of Cauchy matrices are ubiquitous in the theory of Richardson-Gaudin integrability, and an important result by Borchardt [63] showed how the permanent of a Cauchy matrix can be evaluated as a ratio of determinants given by

$$\text{per}[C] = \frac{\det[C * C]}{\det[C]}, \tag{46}$$

with $*$ the Hadamard product, defined as $(A * B)_{ij} = A_{ij}B_{ij}$. The denominator $\det[C]$ can be explicitly evaluated as

$$\det[C] = \frac{\prod_{j<i}(\epsilon_i - \epsilon_j)\prod_{\alpha<\beta}(x_\alpha - x_\beta)}{\prod_{i,\alpha}(\epsilon_i - x_\alpha)}. \tag{47}$$

However, instead of directly evaluating these determinants, it is also possible to rewrite this as

$$\text{per}[C] = \det\left[C^{-1}(C * C)\right] = \det\left[(C * C)C^{-1}\right], \tag{48}$$

as noted in [14]. Because of the known structure of the inverse of a Cauchy matrix, these matrices can be explicitly calculated as

$$C^{-1}(C * C) = D_x J_x D_x^{-1}, \qquad (C * C)C^{-1} = D_\epsilon^{-1} J_\epsilon D_\epsilon, \tag{49}$$

with the $J$-matrices given by (see Appendix B)

$$(J_x)_{\alpha\beta} = \begin{cases} -\sum_i \frac{1}{x_\alpha - \epsilon_i} + \sum_{\kappa \neq \alpha} \frac{1}{x_\alpha - x_\kappa} & \text{if } \alpha = \beta \\ -\frac{1}{x_\alpha - x_\beta} & \text{if } \alpha \neq \beta \end{cases}, \tag{50}$$

$$(J_\epsilon)_{ij} = \begin{cases} \sum_\alpha \frac{1}{\epsilon_i - x_\alpha} - \sum_{k \neq i} \frac{1}{\epsilon_i - \epsilon_k} & \text{if } i = j \\ -\frac{1}{\epsilon_i - \epsilon_j} & \text{if } i \neq j \end{cases}, \tag{51}$$

and the $D$-matrices are those arising in the equation for the inverse (45). Now multiple determinant expressions for the permanent can be used, since

$$\text{per}[C] = \det[J_\epsilon] = \det[J_x]. \tag{52}$$

Because of the product structure of the $J$-matrices, the following identity holds

$$\det[\mathbb{1}_N + J_x] = \det\left[\mathbb{1}_N + C^{-1}(C * C)\right] = \det\left[\mathbb{1}_N + (C * C)C^{-1}\right] = \det[\mathbb{1}_N + J_\epsilon]. \tag{53}$$

Note that so far all involved matrices were $N \times N$ matrices defined in terms of two sets of variables $\{\epsilon_1 \ldots \epsilon_N\}$ and $\{x_1 \ldots x_N\}$. Remarkably, this final expression (53) can immediately be generalized towards matrices of different dimensions, defined in terms of sets with a different number of variables, playing an important role in connecting the inner products to DWPFs. Given two such sets $\{\epsilon_1 \ldots \epsilon_L\}$ and $\{x_1 \ldots x_N\}$, with $L \neq N$, the following also holds

$$\det[\mathbb{1}_N + J_x] = \det[\mathbb{1}_L + J_\epsilon], \tag{54}$$

with $J_x$ an $N \times N$ matrix and $J_\epsilon$ an $L \times L$ matrix defined as in Eq. (50), but now in terms of sets with a different number of variables. This result can be obtained by generalizing Eq. (53) towards non-square matrices and applying Sylvester's determinant identity

$$\det[\mathbb{1}_N + AB] = \det[\mathbb{1}_L + BA], \tag{55}$$

with $A$ an arbitrary $N \times L$ matrix and $B$ an arbitrary $L \times N$ matrix, as commented on in Appendix B.4.

Returning to $N \times N$ matrices, one final property of Cauchy matrices which will be key in relating the eigenvalue-based determinant to Slavnov's determinant, is that

$$2(C * C)^{-1}(C * C * C) = D_x\left[J_x + C^T J_\epsilon^{-1} C\right]D_x^{-1}, \tag{56}$$

as also proven in Appendix B. This can be seen as a higher-order extension of the previous equality

$$C^{-1}(C * C) = D_x J_x D_x^{-1}, \tag{57}$$

and leads to the determinant identity

$$\frac{\det[2(C * C * C)]}{\det[C * C]} = \det\left[J_x + C^T J_\epsilon^{-1} C\right]. \tag{58}$$

# 5  From eigenvalue-based to Slavnov through dual states

## 5.1  Dual states

The crucial element in all eigenvalue-based expressions is the (implicit) existence of dual states for on-shell Bethe states. Any eigenstate of the integrable models under study can be constructed in two different ways: either by creating excitations on top of a vacuum, or by annihilating excitations from a dual vacuum state [12, 21]. The former approach was used in Section 2, where eigenstates were constructed as

$$|v_1 \ldots v_N\rangle = \prod_{a=1}^{N} \left( \sum_{i=1}^{L} \frac{S_i^+}{\epsilon_i - v_a} \right) |\downarrow \ldots \downarrow\rangle, \tag{59}$$

with rapidities $\{v_1 \ldots v_N\}$ satisfying the equations

$$\frac{1}{g} + \frac{1}{2} \sum_{i=1}^{L} \frac{1}{\epsilon_i - v_a} - \sum_{b \neq a}^{N} \frac{1}{v_b - v_a} = 0, \qquad a = 1 \ldots N. \tag{60}$$

However, the latter approach states that eigenstates also have a *dual* representation given by

$$|v_1' \ldots v_{L-N}'\rangle = \prod_{a=1}^{L-N} \left( \sum_{i=1}^{L} \frac{S_i^-}{\epsilon_i - v_a'} \right) |\uparrow \ldots \uparrow\rangle, \tag{61}$$

with the rapidities of this dual state satisfying

$$-\frac{1}{g} + \frac{1}{2} \sum_{i=1}^{L} \frac{1}{\epsilon_i - v_a'} - \sum_{b \neq a}^{L-N} \frac{1}{v_b' - v_a'} = 0, \qquad a = 1 \ldots L-N. \tag{62}$$

Note that these equations are related to the previous ones by changing the sign of $g$, which can be seen as a consequence of the spin-flip symmetry of the conserved charges (16). Because total spin-projection $S^z = \sum_{i=1}^{L} S_i^z$ is a symmetry of the system, this also implies that the same state will in general be described by a different number of rapidities in both representations.

Two such states are eigenstates of a given integrable Hamiltonian by construction, and these can be made to represent the same eigenstate by demanding that their eigenvalues coincide, leading to

$$\sum_{a=1}^{N} \frac{1}{\epsilon_i - v_a} = \sum_{a=1}^{L-N} \frac{1}{\epsilon_i - v_a'} - \frac{2}{g}, \qquad i = 1 \ldots L. \tag{63}$$

While the correspondence between the rapidities of both representations is not intuitive, the correspondence between the eigenvalue-based variables is simple and given by

$$\Lambda_i(\{v_a\}) = \Lambda_i(\{v_a'\}) - \frac{2}{g}, \qquad i = 1 \ldots L. \tag{64}$$

A special case is the $g \to \infty$ limit, where both sets of variables are equal apart from a number of diverging rapidities going to infinity. In this limit, the conserved charges have an additional $su(2)$ total spin symmetry, which is reflected in $\lim_{v \to \infty} S^+(v) \sim \sum_i S_i^+$, as was also observed in [11].

These two different representations of a single state obviously have different normalizations, but it is shown in Appendix C that the ratio of these normalizations is simply given by

$$|v_1 \ldots v_N\rangle = (-1)^N \left( \frac{g}{2} \right)^{L-2N} |v_1' \ldots v_{L-N}'\rangle. \tag{65}$$

This ratio is directly responsible for the appearance of the prefactor in Eq. (29), as will now be shown.

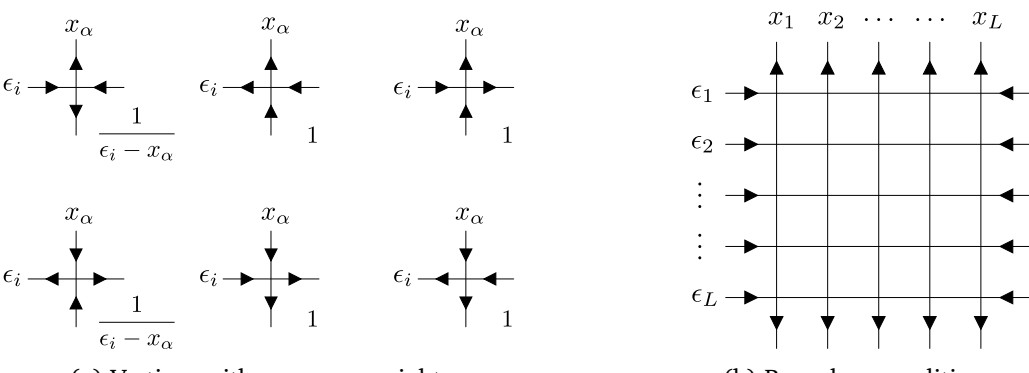

(a) Vertices with non-zero weight.      (b) Boundary conditions.

Figure 2: Graphical representation of the DWPF associated with the Bethe states. The partition function sums over all configurations of vertices in the bulk consistent with the boundary conditions, with the total weight of each configuration given by the product of the weights of the vertices. The weights in the first column follow from $\langle\uparrow|S^+(x_\alpha)|\downarrow\rangle_i$.

## 5.2 Inner products as DWPFs

Since the existence of dual states is guaranteed for on-shell Bethe states, we can always express the inner product in terms of the dual state. The overlap between a (possibly off-shell) state $|w_1\ldots w_N\rangle$ and an on-shell state $|v_1\ldots v_N\rangle$ can now be written as

$$\langle v_1\ldots v_N|w_1\ldots w_N\rangle = (-1)^N\left(\frac{g}{2}\right)^{L-2N}\langle v'_1\ldots v'_{L-N}|w_1\ldots w_N\rangle. \tag{66}$$

The overlap between a dual state and a normal state is simply given by

$$\langle v'_1\ldots v'_{L-N}|w_1\ldots w_N\rangle = \langle\uparrow\ldots\uparrow|\prod_{a=1}^{L-N}\left(\sum_{i=1}^{L}\frac{S_i^+}{\epsilon_i-v'_a}\right)\prod_{b=1}^{N}\left(\sum_{i=1}^{L}\frac{S_i^+}{\epsilon_i-w_b}\right)|\downarrow\ldots\downarrow\rangle \tag{67}$$

$$= \langle\uparrow\ldots\uparrow|\prod_{x_\alpha\in\{v'\}\cup\{w\}}\left(\sum_{i=1}^{L}\frac{S_i^\dagger}{\epsilon_i-x_\alpha}\right)|\downarrow\ldots\downarrow\rangle, \tag{68}$$

which can be interpreted as the overlap of a Bethe state defined by $L$ rapidities $\{x_\alpha\} = \{x_1\ldots x_L\} = \{v'\}\cup\{w\}$ with the dual vacuum $|\uparrow\ldots\uparrow\rangle$,

$$\langle\uparrow\ldots\uparrow|\prod_{\alpha=1}^{L}S^+(x_\alpha)|\downarrow\ldots\downarrow\rangle = \sum_{\sigma\in S_L}\prod_{i=1}^{L}\frac{1}{\epsilon_i-x_{\sigma(i)}}, \tag{69}$$

which is also known as a domain wall boundary partition function (DWPF), as introduced by Korepin [67]. This has a graphical interpretation and is illustrated in Figure 2. For the models at hand, the expression (69) is exactly the definition of the permanent of a matrix

$$\langle\uparrow\ldots\uparrow|\prod_{\alpha=1}^{L}S^+(x_\alpha)|\downarrow\ldots\downarrow\rangle = \text{per}[C], \tag{70}$$

with $C$ the Cauchy matrix defined as

$$C_{i\alpha} = \frac{1}{\epsilon_i-x_\alpha}. \tag{71}$$

Using the results on Cauchy matrices from the previous section (52), this permanent can be rewritten as a determinant of

$$\text{per}[C] = \det[J_\epsilon], \tag{72}$$

with $J_\epsilon$ an $L \times L$ matrix defined as

$$(J_\epsilon)_{ij} = \begin{cases} \sum_{a=1}^{L} \frac{1}{\epsilon_i - x_a} - \sum_{k \neq i}^{L} \frac{1}{\epsilon_i - \epsilon_k} & \text{if } i = j \\ -\frac{1}{\epsilon_i - \epsilon_j} & \text{if } i \neq j \end{cases}. \tag{73}$$

We can now simply reintroduce the rapidities $\{x_1 \ldots x_L\} = \{v'\} \cup \{w\}$ in order to write

$$(J_\epsilon)_{ij} = \begin{cases} \sum_{a=1}^{L-N} \frac{1}{\epsilon_i - v'_a} + \sum_{b=1}^{N} \frac{1}{\epsilon_i - w_b} - \sum_{k \neq i}^{L} \frac{1}{\epsilon_i - \epsilon_k} & \text{if } i = j \\ -\frac{1}{\epsilon_i - \epsilon_j} & \text{if } i \neq j \end{cases}, \tag{74}$$

where, because of the correspondence between the eigenvalues of the original state and the dual state (63), we can write this entirely in terms of the original rapidities to obtain

$$(J_\epsilon)_{ij} = \begin{cases} \frac{2}{g} + \sum_{a=1}^{N} \frac{1}{\epsilon_i - v_a} + \sum_{b=1}^{N} \frac{1}{\epsilon_i - w_b} - \sum_{k \neq i}^{L} \frac{1}{\epsilon_i - \epsilon_k} & \text{if } i = j \\ -\frac{1}{\epsilon_i - \epsilon_j} & \text{if } i \neq j \end{cases}, \tag{75}$$

resulting in the proposed determinant expression (29) for the inner product if we take into account the ratio of normalizations for the original state and the dual state (65). Note that the existence of the dual state was necessary to derive these determinant expressions, but is only implicit in the final results due to the demand that one of the two states in the inner product must be on-shell.

At this point, the first proposed expression (29) has been derived, which is expressed in the eigenvalue-based variables. In order to derive the equivalent expression (30) for the rapidities, the crucial observation is that the matrix (75) has the structure $(\mathbb{1}_L + \cdots)$ after multiplying all matrix elements with $g/2$. This leads to the exact same structure obtained when applying Sylvester's determinant identity in Eq. (54). The involved matrices are now defined in terms of two sets of variables of unequal size, being the $L$ inhomogeneities $\{\epsilon_1 \ldots \epsilon_L\}$ and the $2N$ combined rapidities $\{v_1 \ldots v_N\} \cup \{w_1 \ldots w_N\}$. Applying the relation (54) then connects the determinant of the $L \times L$ matrix $(\mathbb{1}_L + \cdots)$ to that of an $2N \times 2N$ matrix $(\mathbb{1}_{2N} + \cdots)$, equalling the matrix proposed in (30) after correcting for the missing prefactors of $(g/2)$ in the matrix elements by changing the prefactor of the determinant.

Afterwards, the related Gaudin and Izergin-Borchardt determinants follow immediately by taking the appropriate limits.

## 5.3 Reduction to Slavnov's determinant

Starting from the eigenvalue-based inner product, it is possible to rederive Slavnov's determinant expression, obtaining other determinant expressions in the process and shedding some light on the structure of all involved identities.

Firstly, by making using of the Bethe equations (18) the well-known Slavnov's determinant expression (25) can be straightforwardly rewritten as

$$\langle v_1 \ldots v_N | w_1 \ldots w_N \rangle = \frac{\prod_b \prod_a (v_a - w_b)}{\prod_{b<a} (w_b - w_a) \prod_{a<b} (v_b - v_a)} \det S_N(\{v_a\}, \{w_b\}), \tag{76}$$

$$S_N(\{v_a\}, \{w_b\})_{ab} = \frac{1}{(v_a - w_b)^2} \left( \sum_{i=1}^{L} \frac{1}{w_b - \epsilon_i} - 2 \sum_{c \neq a}^{N} \frac{1}{w_b - v_c} - \frac{2}{g} \right). \tag{77}$$

Note that, by rewriting it in this way, the only explicit dependence on the inhomogeneities is through $\sum_i \frac{1}{w_b - \epsilon_i}$, bringing to mind the eigenvalue-based determinants and Eq. (30). The prefactor is the inverse of the determinant of the Cauchy matrix $U$ defined by

$$U_{ab} = \frac{1}{v_a - w_b}. \tag{78}$$

Defining a diagonal matrix $\Lambda$ in order to absorb the dependency of this matrix on the inhomogeneities as

$$\Lambda_{aa} = \sum_{i=1}^{L} \frac{1}{w_a - \epsilon_i} - 2 \sum_{c=1}^{N} \frac{1}{w_a - v_c} - \frac{2}{g}, \tag{79}$$

the matrix in Slavnov's determinant can be decomposed as

$$S_N(\{v_a\}, \{w_b\}) = (U * U)\Lambda - 2(U * U * U). \tag{80}$$

This can be summarized in

$$\langle v_1 \dots v_N | w_1 \dots w_N \rangle = \frac{\det\left[(U * U)\Lambda - 2(U * U * U)\right]}{\det[U]}. \tag{81}$$

We will now work towards this expression starting from the eigenvalue-based expressions using the special properties of Cauchy matrices. The inner product as given in Eq. (30) depends on a $2N \times 2N$ matrix, which can be interpreted as a $2 \times 2$ block matrix of $N \times N$ matrices as

$$\langle v_1 \dots v_N | w_1 \dots w_N \rangle = (-1)^N \det K_{2N}(\{v_a\}, \{w_b\}), \tag{82}$$

with

$$K_{2N}(\{v_a\}, \{w_b\}) = \begin{bmatrix} J_v & -U \\ U^T & J_w - \Lambda \end{bmatrix} \tag{83}$$

and the diagonal matrices are given by

$$(J_v)_{ab} = \begin{cases} \frac{2}{g} - \sum_{i=1}^{L} \frac{1}{v_a - \epsilon_i} + \sum_{c \neq a}^{N} \frac{1}{v_a - v_c} + \sum_{c=1}^{N} \frac{1}{v_a - w_c} & \text{if } a = b \\ -\frac{1}{v_a - v_b} & \text{if } a \neq b \end{cases}, \tag{84}$$

$$(J_w)_{ab} = \begin{cases} -\sum_{c=1}^{N} \frac{1}{w_a - v_c} + \sum_{c \neq a}^{N} \frac{1}{w_a - w_c} & \text{if } a = b \\ -\frac{1}{w_a - w_b} & \text{if } a \neq b \end{cases}. \tag{85}$$

The dependence of $J_w$ on the inhomogeneities can be absorbed in the same diagonal matrix $\Lambda$ as defined for Slavnov's determinant and the off-diagonal matrices are then determined by the same Cauchy matrix defined previously. Using the Richardson-Gaudin equations (18) for $\{v_a\}$ in the diagonal elements of $J_v$ then results in

$$(J_v)_{ab} = \begin{cases} -\sum_{c \neq a}^{N} \frac{1}{v_a - v_c} + \sum_{c=1}^{N} \frac{1}{v_a - w_c} & \text{if } a = b \\ -\frac{1}{v_a - v_b} & \text{if } a \neq b \end{cases}. \tag{86}$$

By rewriting the matrix in this way, all submatrices exhibit the structures previously introduced since $J_v \sim U^{-1}(U * U)$ and $J_w \sim (U * U)U^{-1}$. One final step now consists of relating the determinant of a $2N \times 2N$ matrix to that of an $N \times N$ one. For this, it is possible to evaluate the determinant of a $2 \times 2$ block matrix as

$$\det \begin{bmatrix} A & B \\ C & D \end{bmatrix} = \det(A) \det\left(D - CA^{-1}B\right). \tag{87}$$

Applying this to the equation for the inner product yields

$$\begin{aligned} (-1)^N \det \begin{bmatrix} J_v & -U \\ U^T & J_w - \Lambda \end{bmatrix} &= (-1)^N \det(J_v) \det(J_w - \Lambda + U^T J_v^{-1} U) \\ &= \frac{\det(U * U)}{\det(U)} \det\left[\Lambda - 2(U * U)^{-1}(U * U * U)\right] \\ &= \frac{\det\left[(U * U)\Lambda - 2(U * U * U)\right]}{\det(U)}, \end{aligned} \tag{88}$$

where we have used $\det(J_\nu) = \det(U*U)/\det(U)$ and Eq. (56) to evaluate the inverse of $J_\nu$ as

$$-D_w(J_w - \Lambda + U^T J_\nu^{-1} U)D_w^{-1} = \Lambda - 2(U*U)^{-1}(U*U*U).$$

This corresponds exactly to Slavnov's determinant expression. Alternative ways of calculating the determinant of the block matrix would result in alternative $N \times N$ matrices, but their structure is more involved than that of the Slavnov determinant. Again, the on-shell requirement for one of the two states is crucial, arising here due to the implied existence of a dual state.

In conclusion, there are three ways of evaluating the inner product - by calculating the determinants of $L \times L$, $2N \times 2N$, or $N \times N$ matrices. The structure of all these matrices is intricately related to Cauchy matrices, where the $2N \times 2N$ matrix can be reduced to the $N \times N$ Slavnov determinant, while the $L \times L$ matrix follows from the eigenvalue-based framework. Such results were also obtained in [11] and [68] for the rational six-vertex model, connecting DWPFs with Slavnov determinants, however without invoking the dual representation of Bethe states. Alternatively, similar results were obtained by Kitanine *et al.* for the integrable XXX Heisenberg chain through both a separation of variables and an Algebraic Bethe Ansatz approach [69].

# 6 Extension to hyperbolic models

The presentation thus far was focused on the rational (XXX) Richardson-Gaudin models, because of their clear-cut connection with Cauchy matrices. However, it is possible to extend all results to the hyperbolic (XXZ) Richardson-Gaudin case, for which we will only provide an overview and refer to Appendix D for a detailed derivation. Such models are a generalization of the rational (or XXX) model treated so far, often arising in the context of $p + ip$ superconductivity, where they are known to exhibit a richer phase diagram [32, 70–73].

**Conserved charges and Bethe ansatz**

The class of XXZ Richardson-Gaudin models are defined by a similar set of commuting operators[2] given by

$$R_i = \left(S_i^z + \frac{1}{2}\right) + g\sum_{j\neq i}^{L} \frac{1}{\epsilon_i - \epsilon_j}\left[\sqrt{\epsilon_i \epsilon_j}\left(S_i^+ S_j^- + S_i^- S_j^+\right) + 2\epsilon_i\left(S_i^z S_j^z - \frac{1}{4}\right)\right], \qquad i = 1 \ldots L. \tag{89}$$

The common eigenstates of these operators are given by

$$|\nu_1 \ldots \nu_N\rangle = \prod_{a=1}^{N}\left(\sum_{i=1}^{L} \frac{\sqrt{\epsilon_i}}{\epsilon_i - \nu_a} S_i^+\right)|\downarrow \ldots \downarrow\rangle, \tag{90}$$

with eigenvalues

$$R_i|\nu_1 \ldots \nu_N\rangle = -g\sum_{a=1}^{N} \frac{\epsilon_i}{\epsilon_i - \nu_a}|\nu_1 \ldots \nu_N\rangle, \tag{91}$$

provided the rapidities $\{\nu_1 \ldots \nu_N\}$ satisfy the equations

$$\frac{1 + g^{-1}}{\nu_a} + \sum_{i=1}^{L} \frac{1}{\epsilon_i - \nu_a} - 2\sum_{b\neq a}^{N} \frac{1}{\nu_b - \nu_a} = 0, \qquad \forall a = 1 \ldots N. \tag{92}$$

---

[2]We follow the presentation from [23, 73], which differs from the usual one in the antisymmetry in the last term. However, this allows for a clearer presentation and can be connected to the fact that these models can also be constructed from a non-skew-symmetric $r$-matrix [74–76].

### Eigenvalue-based equations

The conserved charges of these models again satisfy a set of similar quadratic equations [24]

$$R_i^2 = R_i - g\epsilon_i \sum_{j\neq i}^{L} \frac{R_i - R_j}{\epsilon_i - \epsilon_j}, \qquad i = 1\ldots L. \tag{93}$$

Remarkably, the eigenvalues of the conserved charges are determined by eigenvalue-based variables [21] defined in the same way as

$$R_i |v_1 \ldots v_N\rangle = -g\epsilon_i \Lambda_i(\{v_a\}) |v_1 \ldots v_N\rangle, \tag{94}$$

where the eigenvalue-based variables now follow from the quadratic equations

$$\epsilon_i \Lambda_i^2 + \frac{1}{g}\Lambda_i - \sum_{j\neq i}^{L} \frac{\epsilon_i \Lambda_i - \epsilon_j \Lambda_j}{\epsilon_i - \epsilon_j} = 0, \qquad \forall i = 1\ldots L. \tag{95}$$

The connection between this quadratic equation and the original Bethe equations can then be made through the differential equation

$$zP''(z) + \left[1 + g^{-1} + \sum_{i=1}^{L} \frac{z}{\epsilon_i - z}\right] P'(z) - \left[\sum_{i=1}^{L} \frac{\epsilon_i \Lambda_i}{\epsilon_i - z}\right] P(z) = 0. \tag{96}$$

### Inner products

The normalizations of the original and the dual state are now related through (for $L - 2N > 0$)

$$|v_1' \ldots v_{L-N}'\rangle = (-1)^N \frac{\prod_{a=1}^{N} v_a}{\prod_{i=1}^{L} \sqrt{\epsilon_i}} \left[\prod_{k=1}^{L-2N} \left(g^{-1} + 1 - k\right)\right] |v_1 \ldots v_N\rangle. \tag{97}$$

This is a more involved expression compared to the rational model (65), which can be seen as a consequence of the various symmetries and dualities present in this model - for specific values $g^{-1} \in \mathbb{Z}$ the dual wave function vanishes because some of the dual rapidities diverge [77]. Similar terms also arise in the investigation of the phase diagram of these models [70, 72, 78]. The origin of this prefactor is shown in Appendix D and will be further discussed after the presentation of the inner products.

From this, it is possible to obtain a similar set of determinant expressions for inner products in the hyperbolic model, and we find (for on-shell $\{v_a\}$)

$$\langle v_1 \ldots v_N | w_1 \ldots w_N \rangle = (-1)^N \left(\frac{\prod_{i=1}^{L} \epsilon_i}{\prod_{a=1}^{N} v_a}\right) \left[\prod_{k=1}^{L-2N} \left(g^{-1} + 1 - k\right)\right]^{-1} \det J_L(\{v_a\}, \{w_b\}), \tag{98}$$

with $J_L(\{v_a\}, \{w_b\})$ an $L \times L$ matrix given by

$$J_L(\{v_a\}, \{w_b\})_{ij} = \begin{cases} \sum_{a=1}^{N} \frac{1}{\epsilon_i - v_a} + \sum_{b=1}^{N} \frac{1}{\epsilon_i - w_b} + \frac{g^{-1}}{\epsilon_i} - \sum_{k\neq i}^{L} \frac{1}{\epsilon_i - \epsilon_k} & \text{if } i = j \\ -\frac{1}{\epsilon_i - \epsilon_j} & \text{if } i \neq j \end{cases}. \tag{99}$$

The diagonal elements of these matrices again reflect the structure of the eigenvalue-based equations, which can also be used to show the orthogonality of different eigenstates. When calculating the normalization of an on-shell state, this reduces to the Jacobian of the eigenvalue-based equations (95), where all derivatives are now taken w.r.t. $\epsilon_i \Lambda_i$ instead of $\Lambda_i$. Alternatively, there is again a direct equivalence with the determinant of a $2N \times 2N$ matrix defined in

terms of $\{x_1 \ldots x_{2N}\} = \{v_1 \ldots v_N\} \cup \{w_1 \ldots w_N\}$ as

$$\langle v_1 \ldots v_N | w_1 \ldots w_N \rangle = (-1)^N \left( \prod_{b=1}^{N} w_b \right) \det K_{2N}(\{v_a\}, \{w_b\}), \tag{100}$$

with

$$K_{2N}(\{v_a\}, \{w_b\})_{ab} = \begin{cases} -\sum_{i=1}^{L} \frac{1}{x_a - \epsilon_i} + \sum_{c \neq a}^{2N} \frac{1}{x_a - x_c} + \frac{g^{-1}+1}{x_a} & \text{if } a = b \\ -\frac{1}{x_a - x_b} & \text{if } a \neq b \end{cases}, \tag{101}$$

where the diagonal elements now clearly reflect the Bethe equations (92).

The Slavnov determinant for the inner product of an on-shell and an off-shell state in the XXZ model [32] then follows from evaluating this matrix as a block matrix, where all steps in the derivation are fundamentally the same as for the rational XXX model. The resulting expression is given by

$$\langle v_1 \ldots v_N | w_1 \ldots w_N \rangle = \frac{\prod_a \prod_b (v_a - w_b)}{\prod_{a<b}(v_b - v_a) \prod_{b<a}(w_b - w_a)} \det S_N(\{v_a\}, \{w_b\}), \tag{102}$$

with

$$S_N(\{v_a\}, \{w_b\})_{ab} = \frac{w_b}{(v_a - w_b)^2} \left( \sum_{i=1}^{L} \frac{1}{w_b - \epsilon_i} - 2 \sum_{c \neq a}^{N} \frac{1}{w_b - v_c} - \frac{g^{-1}+1}{w_b} \right). \tag{103}$$

## Discussion

As mentioned before, the prefactor in Eq. (97) reflects the various symmetries and dualities present in the hyperbolic model. Since it arises from the ratio of the normalizations of the Bethe state and its dual representation, it can be connected to the behaviour of the dual rapidities. In fact, it is well-established that (dual) rapidities in the XXZ model are allowed to diverge or collapse to zero at specific values of $g^{-1} \in \mathbb{Z}$, and how this reflects additional symmetries in the model at these points [70, 72, 77, 78]. For diverging dual rapidities, the normalization of the dual state will vanish, resulting in a diverging prefactor in Eq. (98). However, this is not a fundamental problem, since this is controllable via a proper normalization of the dual state. It can also easily be checked that this leads to a vanishing determinant, resulting in a well-conditioned finite value of the inner product.

This behaviour can be better understood by making the connection between the dual rapidities and the original rapidities explicit. For the hyperbolic model, the correspondence between the two sets of rapidities is given by [21] as

$$\sum_{a=1}^{N} \frac{1}{\epsilon_i - v_a} + \frac{g^{-1}}{\epsilon_i} = \sum_{a=1}^{L-N} \frac{1}{\epsilon_i - v_a'}, \qquad i = 1 \ldots L. \tag{104}$$

There are now two cases where the dual state vanishes. First, the case where some of the $v_a = 0$ is known to occur only at the so-called Read-Green points [70, 72, 78]. At these points $g^{-1} = -p$, with $p = 1 \ldots N$ the number of vanishing rapidities. The dual rapidities are then given by the $N - p$ non-zero rapidities of the original state supplemented with $L - 2N + p$ diverging rapidities. This can easily be checked as

$$\sum_{a=1}^{L-N} \frac{1}{\epsilon_i - v_a'} = \sum_{a=1}^{N} \frac{1}{\epsilon_i - v_a} - \frac{p}{\epsilon_i} = \sum_{a=1}^{N} \frac{1}{\epsilon_i - v_a} - \frac{g^{-1}}{\epsilon_i}, \tag{105}$$

where the diverging rapidities do not contribute to the summation and the zero rapidities result in a term $p/\epsilon_i$. This hampers a straightforward evaluation of all involved determinants, not

because of the prefactor, but because it necessitates taking the appropriate limit where multiple rapidities coincide. This is a common difficulty in all proposed determinant expressions, requiring a more in-depth analysis falling outside the scope of the current work.

Secondly, the other case where the prefactor might pose trouble is when $g^{-1} = +p$, with $p = 0 \ldots L-2N-1$. Here, all rapidities of the original state are known to be finite and non-zero, and the dual rapidities are given by the $N$ rapidities of the original state, supplemented with $L-2N-p$ diverging dual rapidities and $p$ dual rapidities collapsing to zero. It can again be easily checked that this leads to the correct number of dual rapidities $N+(L-2N-p)+p = L-N$ and

$$\sum_{a=1}^{L-N} \frac{1}{\epsilon_i - v'_a} = \sum_{a=1}^{N} \frac{1}{\epsilon_i - v_a} + \frac{p}{\epsilon_i} = \sum_{a=1}^{N} \frac{1}{\epsilon_i - v_a} + \frac{g^{-1}}{\epsilon_i}. \tag{106}$$

In this case, Eqs. (101) and (102) can be evaluated without difficulties since no rapidities coincide.

The diverging prefactor hence reflects the divergence of a subset of dual rapidities at specific and predictable values of $g^{-1}$ where the XXZ model exhibits an additional symmetry [77], allowing us to relate the dual rapidities to the original rapidities.

# 7 Conclusion

Integrable Richardson-Gaudin models can be investigated in two distinct ways - either by solving the Bethe equations for the rapidities, or by solving a set of quadratic Bethe equations for the eigenvalues of the conserved charges. Similarly, inner products of eigenstates of these integrable models can be expressed in two distinct ways, as determinants of matrices depending on either the rapidities or the eigenvalues. As a direct consequence, the normalization of such eigenstates can be calculated as the determinant of either the Jacobian of the Bethe equations or the Jacobian of the quadratic Bethe equations.

In this work, we investigated the connection between both approaches and presented two classes of determinant expressions for the inner product between on-shell and off-shell states in these integrable models. Slavnov's determinant is then obtained as a special case of one of the two classes of determinants. The crucial element in this connection is the existence of two distinct representations for each eigenstate, allowing inner products to be recast in domain wall boundary partition functions, and the structure of all involved matrices in terms of Cauchy matrices, which has been made explicit here.

# Acknowledgements

P.W.C. acknowledges support from a Ph.D. fellowship and a travel grant for a long stay abroad at the University of Amsterdam from the Research Foundation Flanders (FWO Vlaanderen). P.W.C. thanks J.-S. Caux for valuable discussions and for his hospitality at the University of Amsterdam.

# A Orthogonality of Bethe states

Starting from Eq. (28) or Eq. (30), the orthogonality of different Bethe states can easily be shown. Suppose we have two non-equal on-shell states $\{v_a\}$ and $\{w_b\}$. Then it is straightfor-

ward to show that the rows of (29) are linearly dependent.

$$\sum_{i=1}^{L}[\Lambda_i(\{v_a\}) - \Lambda_i(\{w_b\})]J_L(\{v_a\},\{w_b\})_{ij}$$

$$= \left[\Lambda_j(\{v_a\})^2 + \frac{2}{g}\Lambda_j(\{v_a\}) - \sum_{i\neq j}^{L}\frac{\Lambda_j(\{v_a\}) - \Lambda_i(\{v_a\})}{\epsilon_j - \epsilon_i}\right]$$

$$- \left[\Lambda_j(\{w_b\})^2 + \frac{2}{g}\Lambda_j(\{w_b\}) - \sum_{i\neq j}^{L}\frac{\Lambda_j(\{w_b\}) - \Lambda_i(\{w_b\})}{\epsilon_j - \epsilon_i}\right] = 0, \qquad (107)$$

which vanish since these are exactly the eigenvalue-based equations satisfied by on-shell states (21). The linear dependence of the rows implies that the determinant of this matrix also vanishes, proving the orthogonality of different eigenstates

The orthogonality also follows from the similarity of the diagonal elements of (30) to the Bethe equations (18). Taking $\{x_1 \ldots x_N\} = \{v_1 \ldots v_N\}$ and $\{x_{N+1} \ldots x_{2N}\} = \{w_1 \ldots w_N\}$, this results in

$$\sum_{c=1}^{N}K_{2N}(\{v_a\},\{w_b\})_{cb} - \sum_{c=N+1}^{2N}K_{2N}(\{v_a\},\{w_b\})_{cb}$$

$$= \begin{cases} \frac{2}{g} + \sum_{i=1}^{L}\frac{1}{\epsilon_i - v_b} - 2\sum_{c\neq b}^{N}\frac{1}{v_c - v_b} = 0 & \text{if } b = 1 \ldots N, \\ \frac{2}{g} + \sum_{i=1}^{L}\frac{1}{\epsilon_i - w_b} - 2\sum_{c\neq b}^{N}\frac{1}{w_c - w_b} = 0 & \text{if } b = N+1 \ldots 2N, \end{cases} \qquad (108)$$

where we have identified $w_b$ and $w_{N+b}$ in the second line. Since these are exactly the Bethe equations (18), the rows are again linearly dependent, leading to a vanishing determinant and orthogonal eigenstates.

# B  Properties of Cauchy matrices

## B.1  Inverse of Cauchy matrices

In this Appendix, proofs will be provided for the properties of Cauchy matrices used throughout the main text. For this, the crucial element is the explicit expression for the inverse of a Cauchy matrix. Starting from a Cauchy matrix defined by two sets of variables $\{\epsilon_1 \ldots \epsilon_N\}$ and $\{x_1 \ldots x_N\}$ as

$$C_{i\alpha} = \frac{1}{\epsilon_i - x_\alpha}, \qquad (109)$$

the inverse is given by (see e.g. [66])

$$\left[C^{-1}\right]_{\alpha i} = (\epsilon_i - x_\alpha)\frac{\prod_{k\neq i}(x_\alpha - \epsilon_k)\prod_{\beta\neq\alpha}(\epsilon_i - x_\beta)}{\prod_{k\neq i}(\epsilon_i - \epsilon_k)\prod_{\beta\neq\alpha}(x_\alpha - x_\beta)} = -\frac{1}{\epsilon_i - x_\alpha}\frac{p(x_\alpha)q(\epsilon_i)}{p'(\epsilon_i)q'(x_\alpha)}, \qquad (110)$$

with $p(x) = \prod_i(x - \epsilon_i)$ and $q(x) = \prod_\alpha(x - x_\alpha)$. Explicitly writing out the action of the inverse on the Cauchy matrix results in the identity

$$\sum_\alpha \frac{\epsilon_i - x_\alpha}{\epsilon_j - x_\alpha}\frac{\prod_{k\neq i}(x_\alpha - \epsilon_k)\prod_{\beta\neq\alpha}(\epsilon_i - x_\beta)}{\prod_{k\neq i}(\epsilon_i - \epsilon_k)\prod_{\beta\neq\alpha}(x_\alpha - x_\beta)} = \delta_{ij}, \qquad (111)$$

which is a necessary result for the following. Note that this equality also holds if the set $\{\epsilon_i\}$ contains less variables than the set $\{x_\alpha\}$, since this can be obtained by simply sending an appropriate amount of $\epsilon_i$ to infinity.

## B.2 First order

In the notation of Section 4, the matrix elements of $(C * C)C^{-1}$ can be explicitly calculated. The off-diagonal elements $(i \neq j)$ result in

$$
\left[(C * C)C^{-1}\right]_{ij} = \sum_\alpha \frac{1}{(\epsilon_i - x_\alpha)^2} \frac{1}{(x_\alpha - \epsilon_j)} \frac{p(x_\alpha)q(\epsilon_j)}{p'(\epsilon_j)q'(x_\alpha)} \tag{112}
$$

$$
= \frac{1}{\epsilon_i - \epsilon_j} \sum_\alpha \left[ \frac{1}{(\epsilon_i - x_\alpha)^2} + \frac{1}{(\epsilon_i - x_\alpha)(x_\alpha - \epsilon_j)} \right] \frac{p(x_\alpha)q(\epsilon_j)}{p'(\epsilon_j)q'(x_\alpha)} \tag{113}
$$

$$
= \frac{1}{\epsilon_i - \epsilon_j} \left[ \sum_\alpha \frac{1}{(\epsilon_i - x_\alpha)^2} \frac{p(x_\alpha)q(\epsilon_i)}{p'(\epsilon_i)q'(x_\alpha)} \right] \frac{p'(\epsilon_i)q(\epsilon_j)}{p'(\epsilon_j)q(\epsilon_i)} \tag{114}
$$

$$
= -\frac{1}{\epsilon_i - \epsilon_j} \frac{p'(\epsilon_i)q(\epsilon_j)}{p'(\epsilon_j)q(\epsilon_i)}. \tag{115}
$$

Here, the properties of the inverse as written out in Eq. (111) have been used twice in order to evaluate the summations, first for a vanishing summation in Eq. (113) to Eq. (114), making use of $i \neq j$, and later in order to evaluate the non-zero summation when moving from Eq. (114) to Eq. (115). The diagonal elements can similarly be calculated as

$$
\left[(C * C)C^{-1}\right]_{ii} = \sum_\alpha \frac{1}{\epsilon_i - x_\alpha} \frac{\prod_{k \neq i}(x_\alpha - \epsilon_k)\prod_{\beta \neq \alpha}(\epsilon_i - x_\beta)}{\prod_{k \neq i}(\epsilon_i - \epsilon_k)\prod_{\beta \neq \alpha}(x_\alpha - x_\beta)}. \tag{116}
$$

This can be simplified by performing a partial fraction decomposition in $\epsilon_i$, which has single poles in $x_\alpha$ and $\epsilon_j$ $(j \neq i)$. This results in

$$
\left[(C * C)C^{-1}\right]_{ii} = \sum_\alpha \frac{1}{\epsilon_i - x_\alpha} \left[ \frac{\prod_{k \neq i}(x_\alpha - \epsilon_k)\prod_{\beta \neq \alpha}(x_\alpha - x_\beta)}{\prod_{k \neq i}(x_\alpha - \epsilon_k)\prod_{\beta \neq \alpha}(x_\alpha - x_\beta)} \right]
$$
$$
- \sum_{j \neq i} \frac{1}{\epsilon_i - \epsilon_j} \left[ \sum_\alpha \frac{\prod_{k \neq i,j}(x_\alpha - \epsilon_k)\prod_{\beta \neq \alpha}(\epsilon_j - x_\beta)}{\prod_{k \neq i,j}(\epsilon_j - \epsilon_k)\prod_{\beta \neq \alpha}(x_\alpha - x_\beta)} \right] \tag{117}
$$

$$
= \sum_\alpha \frac{1}{\epsilon_i - x_\alpha} - \sum_{j \neq i} \frac{1}{\epsilon_i - \epsilon_j}, \tag{118}
$$

where again only the properties of the inverse of the Cauchy matrix as in Eq. (111) have been used to evaluate the single summation. This then leads to

$$
(C * C)C^{-1} = D_\epsilon^{-1} J_\epsilon D_\epsilon, \tag{119}
$$

with

$$
(J_\epsilon)_{ij} = \begin{cases} \sum_\alpha \frac{1}{\epsilon_i - x_\alpha} - \sum_{k \neq i} \frac{1}{\epsilon_i - \epsilon_k} & \text{if } i = j \\ -\frac{1}{\epsilon_i - \epsilon_j} & \text{if } i \neq j \end{cases}, \tag{120}
$$

and $D_\epsilon$ a diagonal matrix with $(D_\epsilon)_{ii} = q(\epsilon_i)/p'(\epsilon_i)$. Equivalently, changing the order of the product leads to

$$
C^{-1}(C * C) = D_x J_x^{-1} D_x^{-1}, \tag{121}
$$

with

$$
(J_x)_{\alpha\beta} = \begin{cases} -\sum_i \frac{1}{x_\alpha - \epsilon_i} + \sum_{\kappa \neq \alpha} \frac{1}{x_\alpha - x_\kappa} & \text{if } \alpha = \beta \\ -\frac{1}{x_\alpha - x_\beta} & \text{if } \alpha \neq \beta \end{cases}, \tag{122}
$$

and $D_x$ a diagonal matrix with $(D_x)_{\alpha\alpha} = p(x_\alpha)/q'(x_\alpha)$.

## B.3 Second order

The first-order relations (52) can be extended to higher-order Hadamard products, where the relevant expression for our purpose is

$$2(C * C)^{-1}(C * C * C) = D_x \left[ J_x + C^T J_\epsilon^{-1} C \right] D_x^{-1}. \tag{123}$$

By using the definitions of $J_x$ and $J_\epsilon$, relating the inverse of $C$ to its transposed (45), and multiplying with $C * C$, this can again be expressed purely in terms of Cauchy matrices and diagonal matrices as

$$2(C * C * C) = (C * C)C^{-1}(C * C) - D_\epsilon^{-1} C D_x^{-1}. \tag{124}$$

So, written out explicitly, the equality to be proven is

$$\frac{2}{(\epsilon_i - x_\alpha)^3} = -\frac{p'(\epsilon_i)q'(x_\alpha)}{q(\epsilon_i)p(x_\alpha)}\frac{1}{\epsilon_i - x_\alpha} - \sum_{j \neq i}\frac{p'(\epsilon_i)q(\epsilon_j)}{q(\epsilon_i)p'(\epsilon_j)}\frac{1}{\epsilon_i - \epsilon_j}\frac{1}{(\epsilon_j - x_\alpha)^2}$$

$$-\left[\sum_{j \neq i}\frac{1}{\epsilon_i - \epsilon_j} - \sum_\beta \frac{1}{\epsilon_i - x_\beta}\right]\frac{1}{(\epsilon_i - x_\alpha)^2}, \tag{125}$$

where the previously obtained expression for $(C * C)C^{-1}$ has been inserted. The second term on the right-hand side can be expanded as

$$\frac{1}{\epsilon_i - \epsilon_j}\frac{1}{(\epsilon_j - x_\alpha)^2} = \frac{1}{\epsilon_i - x_\alpha}\frac{1}{(\epsilon_j - x_\alpha)^2} + \frac{1}{(\epsilon_i - x_\alpha)^2}\left[\frac{1}{\epsilon_i - \epsilon_j} + \frac{1}{\epsilon_j - x_\alpha}\right], \tag{126}$$

simplifying this summation to

$$\sum_{j \neq i}\frac{p'(\epsilon_i)q(\epsilon_j)}{q(\epsilon_i)p'(\epsilon_j)}\frac{1}{\epsilon_i - \epsilon_j}\frac{1}{(\epsilon_j - x_\alpha)^2} = \frac{1}{\epsilon_i - x_\alpha}\sum_{j \neq i}\left[\frac{1}{(\epsilon_j - x_\alpha)^2}\frac{p'(\epsilon_i)q(\epsilon_j)}{q(\epsilon_i)p'(\epsilon_j)}\right]$$

$$+ \frac{1}{(\epsilon_i - x_\alpha)^2}\sum_{j \neq i}\left[\frac{1}{\epsilon_i - \epsilon_j} + \frac{1}{\epsilon_j - x_\alpha}\right]\frac{p'(\epsilon_i)q(\epsilon_j)}{q(\epsilon_i)p'(\epsilon_j)}$$

$$= \frac{1}{\epsilon_i - x_\alpha}\frac{p'(\epsilon_i)q'(x_\alpha)}{q(\epsilon_i)p(x_\alpha)} - \frac{1}{(\epsilon_i - x_\alpha)^3}$$

$$+ \frac{1}{(\epsilon_i - x_\alpha)^2}\sum_{j \neq i}\left[\frac{1}{\epsilon_i - \epsilon_j} + \frac{1}{\epsilon_j - x_\alpha}\right]\frac{p'(\epsilon_i)q(\epsilon_j)}{q(\epsilon_i)p'(\epsilon_j)}, \tag{127}$$

where again only the properties of the inverse of the Cauchy matrix have been used. Plugging this into the original equation and multiplying with $(\epsilon_i - x_\alpha)^2$, the equality to be proven is

$$\sum_{j \neq i}\left[\frac{1}{\epsilon_i - \epsilon_j} + \frac{1}{\epsilon_j - x_\alpha}\right]\frac{p'(\epsilon_i)q(\epsilon_j)}{q(\epsilon_i)p'(\epsilon_j)} = \sum_{\beta \neq \alpha}\frac{1}{\epsilon_i - x_\beta} - \sum_{j \neq i}\frac{1}{\epsilon_i - \epsilon_j}. \tag{128}$$

This can be done by doing a partial fraction expansion of the left-hand side in $\epsilon_i$, which has single poles in $\epsilon_i = x_\beta$, $\forall \beta$ and $\epsilon_i = \epsilon_j$, $j \neq i$. This calculation is highly reminiscent of the one used in the previous identities, resulting in

$$\sum_{j \neq i}\left[\frac{1}{\epsilon_i - \epsilon_j} + \frac{1}{\epsilon_j - x_\alpha}\right]\frac{p'(\epsilon_i)q(\epsilon_j)}{q(\epsilon_i)p'(\epsilon_j)} = \sum_\beta \frac{1}{\epsilon_i - x_\beta}\left[1 - \delta_{\alpha\beta}\right] - \sum_{j \neq i}\frac{1}{\epsilon_i - \epsilon_j}$$

$$= \sum_{\beta \neq \alpha}\frac{1}{\epsilon_i - x_\beta} - \sum_{j \neq i}\frac{1}{\epsilon_i - \epsilon_j}, \tag{129}$$

thus proving the identity.

### B.4 Sylvester's determinant identity

In Section 4, it was already mentioned how the factorization of the $J$-matrices leads to the determinant identity

$$\det[\mathbb{1}_N + J_x] = \det[\mathbb{1}_N + J_\epsilon], \tag{130}$$

with $J_x$ and $J_\epsilon$ two $N \times N$ matrices defined according to Eqs. (120) and (122) in terms of two sets of variables $\{\epsilon_1 \dots \epsilon_N\}$ and $\{x_1 \dots x_N\}$. This can now be generalized towards an $L \times L$ matrix $J_\epsilon$ and an $N \times N$ matrix $J_x$ defined in terms of differently-sized sets $\{\epsilon_1 \dots \epsilon_L\}$ and $\{x_1 \dots x_N\}$, leading to

$$\det[\mathbb{1}_N + J_x] = \det[\mathbb{1}_L + J_\epsilon]. \tag{131}$$

This can be done in two ways. The first is by generalizing the factorization properties (119) and (121) towards non-square Cauchy matrices, where $C^{-1}$ is no longer the inverse, but can still be explicitly defined through Eq. (110). The calculation here is the exact same one as for square matrices, since all involved identities hold for differently-sized sets of variables.

Another way is by starting from two sets of equal size, e.g. $L$ if $L > N$, and taking the limit where the variables $\{x_{N+1} \dots x_L\}$ go to infinity as $x_{N+1} \ll x_{N+2} \ll \cdots \ll x_L$. These then drop out of the summation in the diagonal elements, their off-diagonal elements vanish, and the matrix identity for equal sizes reduces to

$$\det[\mathbb{1}_L + J_\epsilon] = \det \begin{bmatrix} \mathbb{1}_{L-N} & 0 \\ 0 & \mathbb{1}_N + J_x \end{bmatrix} = \det[\mathbb{1}_N + J_x]. \tag{132}$$

## C Normalizations of the normal and the dual state

From the identities presented in Appendix B, it is possible to derive the ratio of the normalizations of a dual Bethe state and a regular Bethe state. Since these are both eigenstates, we already know that they describe the same state, but with different normalizations. We now calculate the ratio of the normalization of these states by taking the overlap of both states with an (arbitrary) reference state, and show how it leads to our proposed expression (65).

To reiterate, a Bethe ansatz eigenstate for the Richardson-Gaudin models can be written in two representations as

$$|\{v_a\}\rangle = \prod_{a=1}^{N} \left( \sum_{i=1}^{L} \frac{S_i^+}{\epsilon_i - v_a} \right) |\downarrow \dots \downarrow\rangle, \qquad |\{v_a'\}\rangle = \prod_{a=1}^{L-N} \left( \sum_{i=1}^{L} \frac{S_i^-}{\epsilon_i - v_a'} \right) |\uparrow \dots \uparrow\rangle, \tag{133}$$

with the (dual) rapidities satisfying the Richardson-Gaudin equations

$$\frac{1}{g} + \frac{1}{2} \sum_{i=1}^{L} \frac{1}{\epsilon_i - v_a} - \sum_{b \neq a}^{N} \frac{1}{v_b - v_a} = 0, \qquad a = 1 \dots N, \tag{134}$$

$$-\frac{1}{g} + \frac{1}{2} \sum_{i=1}^{L} \frac{1}{\epsilon_i - v_a'} - \sum_{b \neq a}^{L-N} \frac{1}{v_b' - v_a'} = 0, \qquad a = 1 \dots L-N, \tag{135}$$

describing the same eigenstate if these variables are coupled through

$$\sum_{a=1}^{N} \frac{1}{\epsilon_i - v_a} + \frac{2}{g} = \sum_{a=1}^{L-N} \frac{1}{\epsilon_i - v_a'}, \qquad i = 1 \dots L. \tag{136}$$

The crucial element in this proof is the existence of eigenvalue-based expressions for the inner product of a Bethe state and a reference state [12]. Defining a reference state from a set

of occupied levels can again be done in two ways

$$|\{i_{occ}\}\rangle = \prod_{i \in \{i_{occ}\}} S_i^+ |\downarrow \ldots \downarrow\rangle = \prod_{i \notin \{i_{occ}\}} S_i^- |\uparrow \ldots \uparrow\rangle, \tag{137}$$

where both states are already normalized. Then the overlap of the original Bethe state with this reference state is given by

$$\langle\{i_{occ}\}|\{v_a\}\rangle = \det J_N(\{v_a\}, \{i_{occ}\}), \tag{138}$$

with $J_N(\{v_a\}, \{i_{occ}\})$ an $N \times N$ matrix defined as

$$J_N(\{v_a\}, \{i_{occ}\})_{ij} = \begin{cases} \sum_{a=1}^{N} \frac{1}{\epsilon_i - v_a} - \sum_{\substack{k \in \{i_{occ}\} \\ k \neq i}} \frac{1}{\epsilon_i - \epsilon_k} & \text{if } i = j \\ -\frac{1}{\epsilon_i - \epsilon_j} & \text{if } i \neq j \end{cases}, \qquad i, j \in \{i_{occ}\}. \tag{139}$$

Alternatively, the overlap of the dual state with the same reference state can be written as

$$\langle\{i_{occ}\}|\{v_a'\}\rangle = \langle\downarrow \ldots \downarrow| \left( \prod_{i \in \{i_{occ}\}} S_i^- \right) \left( \prod_{a=1}^{L-N} S^-(v_a') \right) |\uparrow \ldots \uparrow\rangle$$

$$= \langle\downarrow \ldots \downarrow| \left( \prod_{a=1}^{L-N} S^-(v_a') \right) \left( \prod_{i \in \{i_{occ}\}} S_i^- \right) |\uparrow \ldots \uparrow\rangle$$

$$= \langle\downarrow \ldots \downarrow| \left( \prod_{a=1}^{L-N} S^-(v_a') \right) \left( \prod_{i \notin \{i_{occ}\}} S_i^+ \right) |\downarrow \ldots \downarrow\rangle. \tag{140}$$

This final expression is the complex conjugate of the inner product of a regular Bethe state defined by the dual rapidities and a different reference state. Because all eigenvalue-based variables are real, these inner products are always real, and we can use the same determinant expressions to write

$$\langle\{i_{occ}\}|\{v_a'\}\rangle = \det J_{L-N}(\{v_a'\}, \{i | i \notin \{i_{occ}\}\}), \tag{141}$$

with $J_{L-N}(\{v_a'\}, \{i | i \notin \{i_{occ}\}\})$ an $(L-N) \times (L-N)$ matrix defined as

$$J_{L-N}(\{v_a'\}, \{i | i \notin \{i_{occ}\}\})_{ij} = \begin{cases} \sum_{a=1}^{L-N} \frac{1}{\epsilon_i - v_a'} - \sum_{\substack{k \notin \{i_{occ}\} \\ k \neq i}} \frac{1}{\epsilon_i - \epsilon_k} & \text{if } i = j \\ -\frac{1}{\epsilon_i - \epsilon_j} & \text{if } i \neq j \end{cases}, \qquad i, j \notin \{i_{occ}\}. \tag{142}$$

Because this only depends on the dual rapidities through the eigenvalue-based variables in the diagonal elements, the correspondence (136) can be used to express this determinant in the rapidities of the original state as

$$\langle\{i_{occ}\}|\{v_a'\}\rangle = \det\left[ \frac{2}{g} + J_{L-N}(\{v_a\}, \{i | i \notin \{i_{occ}\}\}) \right]$$

$$= \left( \frac{2}{g} \right)^{L-N} \det\left[ \mathbb{1} + \frac{g}{2} J_{L-N}(\{v_a\}, \{i | i \notin \{i_{occ}\}\}) \right]. \tag{143}$$

Now Sylvester's determinant identity can be used, as explained in Section 4, to exchange the role of rapidities and inhomogeneities and obtain

$$\langle\{i_{occ}\}|\{v_a'\}\rangle = \left( \frac{2}{g} \right)^{L-N} \det\left[ \mathbb{1} + \frac{g}{2} K_N(\{v_a\}, \{i | i \notin \{i_{occ}\}\}) \right], \tag{144}$$

with $K_N(\{v_a\}, \{i | i \notin \{i_{occ}\}\})$ an $N \times N$ matrix defined as

$$K_N(\{v_a\}, \{i | i \notin \{i_{occ}\}\})_{ab} = \begin{cases} -\sum_{i \notin \{i_{occ}\}} \frac{1}{v_a - \epsilon_i} + \sum_{c \neq a}^{N} \frac{1}{v_a - v_c} & \text{if } a = b \\ -\frac{1}{v_a - v_b} & \text{if } a \neq b \end{cases}. \tag{145}$$

Here, it is important to note the similarity between the diagonal elements of this matrix and the Richardson-Gaudin equations. By slightly rewriting the Richardson-Gaudin equations, these diagonal elements can be rewritten as

$$1 - \frac{g}{2} \sum_{i \notin \{i_{occ}\}} \frac{1}{v_a - \epsilon_i} + \frac{g}{2} \sum_{c \neq a}^{N} \frac{1}{v_a - v_c} = \frac{g}{2} \sum_{i \in \{i_{occ}\}} \frac{1}{v_a - \epsilon_i} - \frac{g}{2} \sum_{c \neq a}^{N} \frac{1}{v_a - v_c}, \tag{146}$$

resulting in

$$\langle \{i_{occ}\} | \{v'_a\} \rangle = (-1)^N \left( \frac{2}{g} \right)^{L-2N} \det K_N(\{v_a\}, \{i | i \in \{i_{occ}\}\}). \tag{147}$$

Both $g/2$ and the minus sign in the diagonal elements have been absorbed in a prefactor, since multiplying all rows with $-1$ and taking the transpose of the matrix returns the original matrix but with the sign of the diagonal elements exchanged. The roles of both variables can again be exchanged in this final expression to obtain the original equality

$$\langle \{i_{occ}\} | \{v'_a\} \rangle = (-1)^N \left( \frac{2}{g} \right)^{L-2N} \langle \{i_{occ}\} | \{v_a\} \rangle. \tag{148}$$

Since the set of reference states forms a complete basis and this equality holds for all such states, this yields the proposed expression (65) as

$$|\{v'_a\}\rangle = (-1)^N \left( \frac{2}{g} \right)^{L-2N} |\{v_a\}\rangle. \tag{149}$$

# D   Results for the hyperbolic model

## Applying the matrix determinant lemma

All derivations for the hyperbolic model are essentially the same as those for the rational model, except for one additional result. From the matrix determinant lemma, it can be shown using the same arguments as in Appendix B that

$$\left( \prod_{i=1}^{L} \epsilon_i \right) \det \left[ \frac{G-1}{2\epsilon} + J_\epsilon \right] = \left( \prod_{\alpha=1}^{L} x_\alpha \right) \det \left[ \frac{G+1}{2x} + J_x \right], \tag{150}$$

for two sets of equal size $\{\epsilon_1 \ldots \epsilon_L\}$ and $\{x_1 \ldots x_L\}$ and arbitrary $G \in \mathbb{C}$, where we again have $J_\epsilon$ an $L \times L$ as an matrix defined as

$$(J_\epsilon)_{ij} = \begin{cases} \sum_{\alpha=1}^{L} \frac{1}{\epsilon_i - x_\alpha} - \sum_{k \neq i}^{L} \frac{1}{\epsilon_i - \epsilon_k} & \text{if } i = j \\ -\frac{1}{\epsilon_i - \epsilon_j} & \text{if } i \neq j \end{cases}, \tag{151}$$

and $J_x$ also an $L \times L$ matrix defined as

$$(J_x)_{\alpha\beta} = \begin{cases} -\sum_{i=1}^{L} \frac{1}{x_\alpha - \epsilon_i} + \sum_{\kappa \neq \alpha}^{L} \frac{1}{x_\alpha - x_\kappa} & \text{if } \alpha = \beta \\ -\frac{1}{x_\alpha - x_\beta} & \text{if } \alpha \neq \beta \end{cases}. \tag{152}$$

In this expression $1/\epsilon$ and $1/x$ are shorthand for $L \times L$ diagonal matrices with diagonal elements $1/\epsilon_i$ and $1/x_\alpha$ respectively. This follows from the version of the matrix determinant lemma relating

$$\det\left(A + UV^T\right) = \det(\mathbb{1} + V^T A^{-1} U)\det(A), \tag{153}$$

with $A$ an invertible $n \times n$ matrix and $U, V$ are $n \times m$ matrices. This reduces to Sylvester's determinant identity in the special case $A = \mathbb{1}$, and here it is applied by setting $A = 1/\epsilon$, $U = C * C$ and $V^T = C^{-1}$. Since $A$ is a diagonal matrix, both its inverse and its determinant can be explicitly calculated, and all resulting expressions are highly similar to those presented in Appendix B, and will hence not be repeated here.

The limit where the two sets do not contain an equal amount of variables, leading to two matrices of unequal dimensions, can again be obtained by sending the appropriate amount of variables to infinity, similar to the techniques used in [22]. As will be shown explicitly, starting from two sets containing $L$ variables and taking the limit where $L-N$ variables $x_\alpha$ go subsequently to infinity, each diverging variable adds a prefactor due to its appearance in the diagonal elements, giving rise to

$$\left(\prod_{i=1}^{L}\epsilon_i\right)\det\left[\frac{G-1}{2\epsilon} + J_\epsilon\right] = \prod_{k=1}^{L-N}\left(\frac{G+1}{2} - k\right)\left(\prod_{\alpha=1}^{N}x_\alpha\right)\det\left[\frac{G+1}{2x} + J_x\right]. \tag{154}$$

The origin of the prefactors follows from the diagonal elements of the right-hand side and will be derived in full detail, since these are not as trivial as in the rational case. Firstly, starting from Eq. (150), the limit where a single variable is sent to infinity can be obtained. Selecting $x_L \to \infty$ while all other variables remain finite, both sides of Eq. (150) can be evaluated. In the left-hand side, the only dependence on $x_L$ is in the diagonal elements, which reduce to

$$\left[\frac{G-1}{2\epsilon} + J_\epsilon\right]_{ii} = \frac{G-1}{2\epsilon_i} + \sum_{\alpha=1}^{L}\frac{1}{\epsilon_i - x_\alpha} - \sum_{k\neq i}^{L}\frac{1}{\epsilon_i - \epsilon_k} \tag{155}$$

$$= \frac{G-1}{2\epsilon_i} + \sum_{\alpha=1}^{L-1}\frac{1}{\epsilon_i - x_\alpha} - \sum_{k\neq i}^{L}\frac{1}{\epsilon_i - \epsilon_k}, \tag{156}$$

where the limit $x_L \to \infty$ simply removes a single variable from the summation. In order to evaluate the right-hand side, the term $x_L$ from the prefactor can be absorbed in the determinant by multiplying row and column $L$ of the matrix with $x_L^{1/2}$. Taking the limit to infinity, the off-diagonal elements in this row and column will vanish, since these are given by $\pm x_L^{1/2}/(x_L - x_\alpha) \sim x_L^{-1/2} \to 0$. The diagonal element in row/column $L$ is then given by

$$\lim_{x_L\to\infty}\left[\frac{G+1}{2} - \sum_{i=1}^{L}\frac{x_L}{x_L - \epsilon_i} + \sum_{\kappa\neq L}^{L}\frac{x_L}{x_L - x_\kappa}\right] = \frac{G+1}{2} - L + (L-1)$$

$$= \frac{G+1}{2} - 1, \tag{157}$$

while the other diagonal elements will be given by

$$\left[\frac{G+1}{2x} + J_x\right]_{\alpha\alpha} = \frac{G+1}{2x_\alpha} - \sum_{i=1}^{L}\frac{1}{x_\alpha - \epsilon_i} + \sum_{\substack{\kappa\neq\alpha \\ \kappa\neq L}}^{L}\frac{1}{x_\alpha - x_\kappa},$$

$$= \frac{G+1}{2x_\alpha} - \sum_{i=1}^{L}\frac{1}{x_\alpha - \epsilon_i} + \sum_{\beta\neq\alpha}^{L-1}\frac{1}{x_\alpha - x_\beta}, \tag{158}$$

where the term containing $x_L$ again drops out of the summation. Since the off-diagonal elements vanish, the diagonal element will simply contribute a prefactor in the determinant, and the evaluation of the determinant of the $L \times L$ matrix reduces to that of a similar $(L-1) \times (L-1)$ matrix with an additional prefactor $(G+1)/2 - 1$. The effect of multiple diverging variables can be obtained by subsequently sending $x_{L-1}$ to infinity and again absorbing the prefactor $x_{L-1}$ in the matrix. The off-diagonal elements in the row/column $(L-1)$ will again vanish, and the related diagonal element contributes a prefactor

$$\lim_{x_{L-1} \to \infty} \left[ \frac{G+1}{2} - \sum_{i=1}^{L} \frac{x_{L-1}}{x_{L-1} - \epsilon_i} + \sum_{\beta \neq L-1}^{L-1} \frac{x_{L-1}}{x_{L-1} - x_\beta} \right] = \frac{G+1}{2} - L + (L-2)$$
$$= \frac{G+1}{2} - 2. \tag{159}$$

This can be repeated until $N$ variables $\{x_1 \dots x_N\}$ remain, resulting in the proposed expression

$$\left( \prod_{i=1}^{L} \epsilon_i \right) \det \left[ \frac{G-1}{2\epsilon} + J_\epsilon \right] = \prod_{k=1}^{L-N} \left( \frac{G+1}{2} - k \right) \left( \prod_{\alpha=1}^{N} x_\alpha \right) \det \left[ \frac{G+1}{2x} + J_x \right], \tag{160}$$

with $J_\epsilon$ an $L \times L$ matrix again defined as

$$(J_\epsilon)_{ij} = \begin{cases} \sum_{\alpha=1}^{N} \frac{1}{\epsilon_i - x_\alpha} - \sum_{k \neq i}^{L} \frac{1}{\epsilon_i - \epsilon_k} & \text{if } i = j \\ -\frac{1}{\epsilon_i - \epsilon_j} & \text{if } i \neq j \end{cases}, \tag{161}$$

and $J_x$ an $N \times N$ matrix as

$$(J_x)_{\alpha\beta} = \begin{cases} -\sum_{i=1}^{L} \frac{1}{x_\alpha - \epsilon_i} + \sum_{\kappa \neq \alpha}^{N} \frac{1}{x_\alpha - x_\kappa} & \text{if } \alpha = \beta \\ -\frac{1}{x_\alpha - x_\beta} & \text{if } \alpha \neq \beta \end{cases}. \tag{162}$$

**Normalizations of the normal and the dual state**

The derivation from Appendix C can now be repeated for the hyperbolic model, where we will pay special attention to the points where it deviates from the rational model. In the hyperbolic case the Bethe ansatz eigenstate and its dual representation are given by

$$|\{v_a\}\rangle = \prod_{a=1}^{N} \left( \sum_{i=1}^{L} \frac{\sqrt{\epsilon_i}}{\epsilon_i - v_a} S_i^+ \right) |\downarrow \dots \downarrow\rangle, \qquad |\{v_a'\}\rangle = \prod_{a=1}^{L-N} \left( \sum_{i=1}^{L} \frac{\sqrt{\epsilon_i}}{\epsilon_i - v_a'} S_i^- \right) |\uparrow \dots \uparrow\rangle, \tag{163}$$

with the rapidities satisfying the Richardson-Gaudin equations

$$\frac{1 + g^{-1}}{v_a} + \sum_{i=1}^{L} \frac{1}{\epsilon_i - v_a} - 2 \sum_{b \neq a}^{N} \frac{1}{v_b - v_a} = 0, \qquad a = 1 \dots N, \tag{164}$$

$$\frac{1 - g^{-1}}{v_a} + \sum_{i=1}^{L} \frac{1}{\epsilon_i - v_a'} - 2 \sum_{b \neq a}^{L-N} \frac{1}{v_b' - v_a'} = 0, \qquad a = 1 \dots L-N, \tag{165}$$

where the correspondence between both is given by [21]

$$\sum_{a=1}^{N} \frac{1}{\epsilon_i - v_a} + \frac{g^{-1}}{\epsilon_i} = \sum_{a=1}^{L-N} \frac{1}{\epsilon_i - v_a'}, \qquad i = 1 \dots L. \tag{166}$$

A reference state can again be introduced as

$$|\{i_{occ}\}\rangle = \prod_{i\in\{i_{occ}\}} S_i^+ |\downarrow\ldots\downarrow\rangle = \prod_{i\notin\{i_{occ}\}} S_i^- |\uparrow\ldots\uparrow\rangle, \tag{167}$$

and the overlap of the original Bethe state with this reference state is given by [21]

$$\langle\{i_{occ}\}|\{v_a\}\rangle = \sqrt{\prod_{i\in\{i_{occ}\}} \epsilon_i} \det J_N(\{v_a\},\{i_{occ}\}), \tag{168}$$

with $J_N(\{v_a\},\{i_{occ}\})$ an $N\times N$ matrix defined as

$$J_N(\{v_a\},\{i_{occ}\})_{ij} = \begin{cases} \sum_{a=1}^{N} \frac{1}{\epsilon_i-v_a} - \sum_{\substack{k\in\{i_{occ}\}\\k\neq i}} \frac{1}{\epsilon_i-\epsilon_k} & \text{if } i=j \\ -\frac{1}{\epsilon_i-\epsilon_j} & \text{if } i\neq j \end{cases}, \qquad i,j\in\{i_{occ}\}. \tag{169}$$

This is constructed in the exact same way as for the rational model, with only an additional prefactor because of the terms $\sqrt{\epsilon_i}$ in the Bethe state. Similarly, the overlap of the dual state with the same reference state is given by

$$\langle\{i_{occ}\}|\{v_a'\}\rangle = \sqrt{\prod_{i\notin\{i_{occ}\}} \epsilon_i} \det J_{L-N}(\{v_a'\},\{i|i\notin\{i_{occ}\}\}), \tag{170}$$

with $J_{L-N}(\{v_a'\},\{i|i\notin\{i_{occ}\}\})$ an $(L-N)\times(L-N)$ matrix defined as

$$J_{L-N}(\{v_a'\},\{i|i\notin\{i_{occ}\}\})_{ij} = \begin{cases} \sum_{a=1}^{L-N} \frac{1}{\epsilon_i-v_a'} - \sum_{\substack{k\notin\{i_{occ}\}\\k\neq i}} \frac{1}{\epsilon_i-\epsilon_k} & \text{if } i=j \\ -\frac{1}{\epsilon_i-\epsilon_j} & \text{if } i\neq j \end{cases}, \qquad i,j\notin\{i_{occ}\}. \tag{171}$$

The eigenvalue-based variables can again be identified in the diagonal elements, and the correspondence (166) then leads to

$$\langle\{i_{occ}\}|\{v_a'\}\rangle = \sqrt{\prod_{i\notin\{i_{occ}\}} \epsilon_i} \det\left[\frac{g^{-1}}{\epsilon} + J_{L-N}(\{v_a\},\{i|i\notin\{i_{occ}\}\})\right], \tag{172}$$

where $1/\epsilon$ here stands for the diagonal matrix with matrix elements $\epsilon_i, i\notin\{i_{occ}\}$. This is where the matrix determinant lemma (150) comes into play, this time exchanging the role of rapidities and inhomogeneities as

$$\langle\{i_{occ}\}|\{v_a'\}\rangle = \left[\prod_{k=1}^{L-2N}\left(g^{-1}+1-k\right)\right]\frac{\prod_{a=1}^{N} v_a}{\sqrt{\prod_{i\notin\{i_{occ}\}} \epsilon_i}} \det\left[\frac{1+g^{-1}}{v} + K_N(\{v_a\},\{i|i\notin\{i_{occ}\}\})\right], \tag{173}$$

with $K_N(\{v_a\},\{i|i\notin\{i_{occ}\}\})$ an $N\times N$ matrix defined as

$$K_N(\{v_a\},\{i|i\notin\{i_{occ}\}\})_{ab} = \begin{cases} -\sum_{i\notin\{i_{occ}\}} \frac{1}{v_a-\epsilon_i} + \sum_{c\neq a}^{N} \frac{1}{v_a-v_c} & \text{if } a=b \\ -\frac{1}{v_a-v_b} & \text{if } a\neq b \end{cases}, \tag{174}$$

and $1/v$ shorthand for a diagonal matrix with matrix elements $1/v_a$. Remarkably, the diagonal elements again resemble the Richardson-Gaudin equations (164), although now those for the hyperbolic model, and can be rewritten as

$$\frac{1+g^{-1}}{v_a} - \sum_{i\notin\{i_{occ}\}} \frac{1}{v_a-\epsilon_i} + \sum_{c\neq a}^{N} \frac{1}{v_a-v_c} = \sum_{i\in\{i_{occ}\}} \frac{1}{v_a-\epsilon_i} - \sum_{c\neq a}^{N} \frac{1}{v_a-v_c}, \tag{175}$$

leading to

$$\langle\{i_{occ}\}|\{v'_a\}\rangle = (-1)^N \left[\prod_{k=1}^{L-2N}\left(g^{-1}+1-k\right)\right]\frac{\prod_{a=1}^N v_a}{\sqrt{\prod_{i\notin\{i_{occ}\}}\epsilon_i}}\det K_N(\{v_a\},\{i|i\in\{i_{occ}\}\}). \quad (176)$$

As for the rational model, the minus sign in the diagonal elements has been absorbed in the prefactor. As a final step, the roles of both variables can be exchanged by using Sylvester's determinant identity, in order to obtain

$$\langle\{i_{occ}\}|\{v'_a\}\rangle = (-1)^N \left[\prod_{k=1}^{L-2N}\left(g^{-1}+1-k\right)\right]\frac{\prod_{a=1}^N v_a}{\sqrt{\prod_{i=1}^L \epsilon_i}}\langle\{i_{occ}\}|\{v_a\}\rangle, \quad (177)$$

or, equivalently,

$$|\{v'_a\}\rangle = (-1)^N \left[\prod_{k=1}^{L-2N}\left(g^{-1}+1-k\right)\right]\frac{\prod_{a=1}^N v_a}{\sqrt{\prod_{i=1}^L \epsilon_i}}|\{v_a\}\rangle. \quad (178)$$

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
