# Peer review of "Inner products in integrable Richardson-Gaudin models"

_SciPost Physics, doi:SciPost Phys. 3, 028 (2017)_

## Round 1 · Referee Report · Anonymous · 2017-6-30

Strengths

1. Several new, nontrivial and useful identities established.
2. New representations for the scalar products of the Gaudin model
3. Normalisation relation with dual Bethe representation

Weaknesses

1. Insufficient discussion on the trigonometric case
2. Authors skip some parts of the derivation

Report

The authors of the paper ``Inner products in integrable Richardson-Gaudin models'' consider different representations for the scalar products of the rational and trigonometric Gaudin model. They prove an equivalence between different representations for the scalar products, namely the Izergin determinant involving dual rapidities and a suitable modification of the Slavnov determinant. To prove this correspondence the authors propose several useful identities, some of them are quite amazing (identity (53) seems to be completely new and really non-trivial). They also establish a normalisation relation between the initial and dual Bethe states which is always a complicated task. It is a very good paper and without doubt I recommend it for publication, however I think that some improvements are necessary before definitely accepting the paper.

1. The trigonometric (or hyperbolic) case should be better explained. In my opinion the appendix D is too short, the authors should give a detailed derivation of the prefactors in the normalisation formula (94).

2.There should be more discussion on the pathological cases where some prefactors in the normalisation formula (94) are zero. Apparently it leads to divergencies in the scalar product formula (95). The authors should clearly explain what these divergencies mean.

3. For the rational case the authors should give more details (and at least some equations!) for the derivation of the formula (30) in the end of the subsection 5.2 as it is essential for the next subsection.

4. The right citation for the Izergin determinant formula is missing:
A.G. Izergin, Partition function of the six-vertex model in a finite volume. Sov. Phys. Dokl., 32:878--879, 1987.

5. In the end of the section 5 authors rightly mention that ``Such results were also obtained in [11] and [66] for the rational six-vertex model, connecting DWPFs with Slavnov determinants, however without invoking the dual representation of Bethe states." The dual representations of the Bethe states in this framework for the XXX spin chain (with very similar results but using different identities) are discussed in details in:
N. Kitanine, J.M. Maillet, G. Niccoli, and V.Terras, On determinant representations of scalar products and form factors in the SoV approach: the XXX case. J. Phys. A: Math. Theor., 49:104002, 2016. arXiv:1506.02630.

6. In the equation (15) the authors use different normalisation of $S^2(u)$ with respect to the definition (8), it should be clearly stated to avoid confusion.

The paper is well written, most of the the statements are proved in a clear way. It seems that there are very few misprints, I found only one in the equation (111) where in the second line, second product in the numerator $x_\alpha$ should be replaced by $x_\beta$

In conclusion I think that the paper should be published in SciPost after addressing the issues listed above.

Requested changes

1. Give the missing derivations and discussion (points 1.-3. of the report)
2. Add missing references (points 4. and 5. of the report)

  • validity: high
  • significance: good
  • originality: high
  • clarity: high
  • formatting: good
  • grammar: good

Author:  Pieter W. Claeys  on 2017-09-13  [id 170]

(in reply to Report 1 on 2017-06-30)

We would like to thank the referee for his/her detailed reading of the manuscript and for various helpful comments. A point by point reply to the questions/remarks in the report is given in the following.

1 - We have greatly expanded Appendix D, showing the origin of the prefactor by explicitly deriving Eq. (146) and the ratio of the normalizations of both states.

2 - We have added a discussion at the end of Section 6, showing how the prefactor encodes all cases where the rapidities in the dual state diverge, leading to a vanishing normalization ratio. It has also been mentioned where this leads to problems when evaluating the determinant expressions.

3 - We have expanded the derivation of Eq. (30) at the end of the relevant subsection, while also expanding the derivation of Eq. (53), both in the main text and at the end of Appendix B.

4 - We would like to thank the referee for pointing this out, and the reference has been added.

5 - We would also like to thank the referee for pointing us towards this reference. It has been rightly mentioned in the updated main text. However, we avoid mentioning the fact that dual states were also used in this work. These were used in a different way in order to obtain similar determinant identities, and we feel that this might confuse the reader without a detailed discussion, which falls outside the scope of the current paper.

6 - This has been corrected.

We would also like to thank the referee for spotting the typo in Eq. (111).

---

## Round 1 · Referee Report · Anonymous · 2017-7-17

Strengths

1- New determinants expressions for XXX and XXZ Richardson-Gaudin inner products

2- Complete connection established between various such representations

Weaknesses

1 - The intricate structure of the paper makes it sometimes hard to read:

stating results in 3.1, to using them to find new results in 3.2 and 3.3 to making general mathematical statements about Cauchy matrices in 4. Only then in 5 do we prove the originally stated results, and in each and every step, parts of the proofs are relegated to appendices

2- Some of the proof are not described in any mathematical details particularly for XXZ extensions

Report

Very interesting work deriving new determinant representations of inner products. In this work the proof of the explicit ratio of the normalisations between normal and dual representation of the eigenstates form a central basis of their work.

The explicit connection with Slavnov's determinant is established through a thorough study of Cauchy matrices identities.

The work makes valuable advances in the mathematical structures of Richardson-Gaudin models and should therefore be published after the requested changes have been made.

Requested changes

1.
The operators S^2(u) are NOT the Casimir of the GGA since they DO NOT commute with its generators. While they might look like a extension of the SU(2) Casimir, they need to be called by a different name here. Mentioned 4 times : (below eq. 7, below eq. 9, above eq. 11, above eq. 12)

2.
Below eq. 10, it should be mentioned explicitly that the vacuum reference state has to be eigenstate of S^z(u) as well, since it will be used right after the equation to define F(u).

3.
At top of page 7. The sentence “This presents a trade-off.” Is particularly awkward.
It should be either removed or extended in order for the reader to understand what “THIS” actually is and what this trade-off is suppose to be.

4.
It should be pointed out for eq. (27) to (31) that they will be explicitly demonstrated (in section 5.2). Not mentioning this fact leads the reader guessing whether they should be understood already.

5.
Below equation (31) there is a typo which reads “DIAGONEL elements”. Both discussions about the similarities of diagonal elements with either the eigenvalue-based equations and the Bethe equations as well as the following paragraph about the link with the structure of Gaudin matrix are not particularly clear at that point.

The authors should therefore expand a bit on which similarities they want to point out [ especially (28) and (21) do not seem particularly similar to the reader]. Identically, for the next paragraph, the Gaudin matrix have not been presented yet and when they are in the next section (3.2) the similarities are not as striking as the authors claim they are. The remark concerning the explicit connection through the Jacobian will ultimately make the connection clear.

These comments in section 3.1 should therefore either be clarified or simply be delayed to the respective section where the proof will make their meaning clearer.

6. In section 3.2 it is said that coinciding rapidities in eq. (30) leads to the Same Gaudin matrix. A sketch of the proof (or a reference is it was shown in some other paper) should be provided since going from a 2N x 2N matrix with diverging terms to a N x N Gaudin matrix is not a trivial result.

7. Below eq. (52) the authors say “This determinant expression also holds if … may have a different dimension”. I suppose they are only talking about relations (52) but that should be made explicit. They then go and say “This can ALSO be seen as a consequence …”. The ALSO seems to imply that there is another way than sending variables to infinity to demonstrate this fact. If there is another way and then it should be made clear what it is since all of the relations (mainly (48) in this case) so far were built on a Cauchy matrix being a square matrix. If the only way they have to prove it is through the infinity limit, a short discussion of how this limit results in lower dimensionality of (49) or (50) would be useful for the reader.

8. In the first chapter of section 5.1. it is said that “Any eigenstate of an integrable model can be constructed in two different ways …”. It should be made clear that, while it is true for the specific models studied in this work, it is not necessarily the case (at least not trivially) for every integrable model as the sentence seems to imply.

9. Since the dual Bethe equations (58) and correspondence (60) and (61) have been published before, a citation to previous work should be made in section 5.1.

10. The last paragraph of 5.2 starts with a typo “At this pointS”

11. The authors state at the end of section 5.2 that (30) is found by using the Sylvester identity (52). The proof should be made explicit, Since Det[1+Je] = Det[1+Jx], should lead to the LxL matrix Je getting replaced by an LxL matrix Jx built out of the N values of w and the L-N values of v’. The change of dimension going from L to 2N, is not straightforward to see. This is certainly related to point 8 of this report where the discussions of section 4 did not necessarily make the implications on changing dimensionalities as simple to grasp as the authors imply.

12. Citations to earlier work (including the authors’ Read-Green resonances … paper) should be present in the eigenvalue-based equations section of 6, to make it clear that eq. (90), for example, was known previously.

13. The inner product given in (95) is called (by mistake ?), in the sentence above the equation, a Slavnov determinant. So far, a J_L matrix would have been called by the authors an eigenvalue-based expression while they use the Slavnov name for (99).

14. We go from eq. (107) to (108) by having the second term in the bracket of (107) equal to zero (after the sum has been performed of course). It is not obvious why it is the case since the authors say only relation (105) has been used and I think it should be clarified.

15. The 6 points mentioned in Anonymous Report 180, should also be explicitly taken into account.

- This is particularly true concerning the lack of details in Appendix D which makes it hard to track down how the results of this appendix are used in section 6. Explicit references to the equations (and their numbers) for XXX, the replacements to be made, the techniques to be reused, and the parts of the proofs where (52) should be replaced by (141-142) should be made explicit.

- Moreover when addressing the diverging dual rapidities and vanishing wave-function discussed around (94) it should be mention whether this is a fundamental problem or whether those vanishing prefactors are controllable via the proper normalisation of the state.

  • validity: high
  • significance: good
  • originality: high
  • clarity: good
  • formatting: good
  • grammar: good

Author:  Pieter W. Claeys  on 2017-09-13  [id 171]

(in reply to Report 2 on 2017-07-17)

We would like to thank the referee for his/her detailed reading of the manuscript and for various helpful comments. A point by point reply to the questions/remarks in the report is given in the following.

1 - This was on oversight on our part, and has been corrected in the new version.

2 - This has been mentioned explicitly.

3 - As also mentioned in our reply to Anonymous Report 3, we have extended this section in order to better convey the trade-off inherent when choosing either the rapidity-based or the eigenvalue-based methods. We hope this makes our intended meaning more clear.

4 - Such a mention has been added.

5 - The typo has been corrected, and the comments on the similarities of the matrices have been expanded, where we have tried to discuss the overall structure of these matrices and its relation to Gaudin matrices.

6 - We have found multiple ways of deriving the Gaudin matrix from Eq. (30). Since the Gaudin determinant can be derived from the Slavnov determinant, and we show how the Slavnov determinant follows from Eq. (30), we have not added a full proof. The Gaudin matrix can also be constructed by taking the limit where the two sets of rapidities coincide, taking a series expansion of the matrix elements and performing elementary row and column operations leaving the determinant invariant, as has been mentioned in the main text. Although these operations follow naturally from the limit, these are quite cumbersome to write down in detail, so this full proof is again not given.

7 - We have added a discussion on the multiple ways of deriving Eq. (52) at the end of Appendix B and have rewritten the related paragraph in the main text in order to make the results for determinants of matrices of different dimensions more clear.

8 - This sentence has been corrected.

9 - We have added the correct references.

10 - This has been corrected.

11 - See the reply to point 7 of the Report.

12 - We have added the correct references.

13 - We intended to mention that it is possible to derive a Slavnov determinant starting from the eigenvalue-based determinants for the hyperbolic model, but we agree that the sentence as written before was confusing to the reader. This has been corrected.

14 - We have generalized Eq. (105) (Eq. (110) in the new version) in the Appendix, and have made clear when and how it is being used in this derivation.

15 - We have added a discussion at the end of Section 6 and have expanded Appendix D in order to accommodate both points raised by the referee. See also the reply to Anonymous Report 1.

---

## Round 1 · Referee Report · Anonymous · 2017-7-28

Strengths

1- New determinant formulas, some impressive looking, for inner products in Richardson-Gaudin models.
2- Clarifies the connection between the Bethe equation framework and the eigenvalue T-Q like based framework.

Weaknesses

1- Presentation can be improved.
2- Hyperbolic case should be discussed in more depth.

Report

This paper deals with overlaps between off-shell and on-shell in Richardson-Gaudin models. The authors derive two types of determinant formulas for those, show how they are related and how they may be interpreted. Crucial to their analysis are a bunch of identities involving Cauchy matrices, some of which were not known before.

Overall I think this is quite an interesting paper. Most of the results are proven in a quite simple and convincing way, and those will be useful to the community studying such models. I have a few comments, which are listed below. Provided those are addressed, I recommend publication in Scipost.

Requested changes

1- As it is written, the logic of section 4 is not so clear. The author start by considering square matrices, which is fine. However, the discussion after (52) is much too short, and slightly confusing. First, the Sylvester identity (det(1+AB)=det(1+BA)) should be recalled explicitly. Second, it is only really nontrivial in case A and B are not square. Third, (52) becomes confusing since C^{-1} is not an inverse anymore.

I suggest treating the non square case with more care, mentioning explicitly at each step what are the dimensions of the matrices considered, and how the limits are taken.

2- Section 6 (and the corresponding very short appendix D) are quite difficult to read, as the author only explain the differences with the rational case. They should consider expanding the section a bit (or defer other technical steps to an expanded appendix D).

3- In general, the authors should try to quote more the equations they prove, especially when several identities on difference pages are needed. For example after (28), recall the $\Lambda_i$ are given by (22).

Regarding references:

4- For domain wall partition functions, it might be useful to mention that they were introduced in [Korepin, Comm. Math. Phys. 1982]. The determinant formula was first derived in [Izergin, Sov. Phys. Dokl. 1987].
5- Reference 17 was written by Gaudin and translated by Caux.

I noticed a few typos:

6- last paragraph in section 1: "previous expressions" --> "previously known expressions". "extended towards" -->"extended to".
7- page 7: I do not understand what the sentences "This presents a trade-off" and "[...] the differential equation (24) then poses another, intermediate, way" mean.
8- page 8: "diagonel" ->"diagonal".
9- In (105), the top left x_\alpha should be x_\beta.
10- Appendix C: "a dual Bethe state" --> "of a dual Bethe state"

  • validity: high
  • significance: good
  • originality: high
  • clarity: good
  • formatting: excellent
  • grammar: good

Author:  Pieter W. Claeys  on 2017-09-13  [id 172]

(in reply to Report 3 on 2017-07-28)

We would like to thank the referee for his/her detailed reading of the manuscript and for various helpful comments. A point by point reply to the questions/remarks in the report is given in the following.

1 - We relate to the referee's comment, and have tried to make the connection between matrices of different dimensions more clear. Throughout the text, the dimensions of all involved matrices have been made more explicit, especially in the section on Cauchy matrices and in the derivation of Eq. (30), and we have added a subsection to Appendix B discussing the derivation of Eq. (52) in more detail. This subsection also contains more detail on the necessary limits. The Sylvester identity has also been recalled, and we briefly discuss the interpretation of $C^{-1}$ in this case.

2 - We have chosen to retain the overall structure of Section 6, but have greatly expanded Appendix D in order to better present the technical steps of the derivation. At the end of Section 6, we have also added a discussion on the prefactors arising in the hyperbolic model.

3 - This has been taken into account, and the example given by the referee has been addressed.

4 - We would like to thank the referee for pointing this out, and the reference has been added.

5 - The reference has been corrected.

6 - This has been corrected.

7 - We have extended this section in order to better convey the intended meaning. We hope this make it more clear.

8 - This has been corrected.

9 - We would definitely like to thank the referee for spotting this typo, which has been corrected.

10 - This has been corrected.

---

## Round 2 · Referee Report · Anonymous (Referee 3) · 2017-9-25

Report

I believe the authors have successfully addressed all my questions. I recommend publication.

---

## Round 2 · Referee Report · Anonymous (Referee 1) · 2017-9-27

Report

The authors followed all the suggestions from my previous report, the paper can be published in the present form

---

## Round 2 · Referee Report · Anonymous (Referee 2) · 2017-10-2

Report

The requested changes have all been addressed. The submission should therefore be published in SciPost physics.

---

## Round 2 · Author Response

We would like to thank all three referees for their detailed reading of the manuscript, their positive assessment, and for various helpful comments on the structure of the paper. We feel that these have improved the overall quality of the paper, and have provided an extensive overview of the changes in response to the referees' suggestions.

For the authors,
Pieter W. Claeys

---

## Round 2 · List of Changes

Here we provide a list of the most important changes in the current manuscript, and refer to our replies to the reports for a more detailed overview.

  • Appendix D has been extended in order to accommodate the referees' comments and provide a full derivation for the hyperbolic model.
  • A new subsection discussing the results for the hyperbolic model has been added at the end of Section 6.
  • Throughout the text, we pay special attention to the dimensions of the involved matrices when connecting determinants of matrices with different dimensions.
  • The introduction of Section 2.3 has been expanded.
  • The discussion at the end of section 3.1 has been rewritten.
  • The references provided by the referees have been added.
  • Minor changes have also occurred throughout the text following specific remarks/questions by referees, all of which have been listed in our replies to the reports.

---

## Editorial Decision

published